# Cardiovascular and renal effects of apelin in chronic kidney disease: a randomised, double-blind, placebo-controlled, crossover study

Chronic kidney disease (CKD) affects ~10% of the population and cardiovascular disease is its commonest complication. Despite treatment, patient outcomes remain poor and newer therapies are urgently needed. Here, we investigated the systemic and renal effects of apelin in CKD. In a randomized, double-blind, placebo-controlled, crossover study, 24 subjects (12 patients with CKD and 12 matched healthy subjects) received pyroglutamated apelin-13 ([Pyr[1]]apelin-13, 1 nmol/min and 30 nmol/min) or matched placebo on two separate visits. Systemic and renal hemodynamics were monitored throughout. The co-primary endpoints were change in systemic vascular resistance index and renal blood flow. Secondary endpoints were change in blood pressure, cardiac output, pulse wave velocity, glomerular filtration rate, natriuresis, free water clearance and urinary protein excretion. In both health and CKD, 30 nmol/min [Pyr[1]]apelin-13 reduced mean arterial pressure by ~4%, systemic vascular resistance by ~12%, and increased cardiac index by ~10%, compared to placebo (p < 0.05 for all). Both doses of [Pyr[1]]apelin-13 increased renal blood flow by ~15%, natriuresis by ~20% and free water clearance by ~10%, compared to placebo (p < 0.05 for all). In patients with chronic kidney disease only, glomerular filtration rate fell by ~10%, effective filtration fraction by ~5% and proteinuria by ~25% (p < 0.01 for all). Apelin has short-term cardiovascular and renal benefits in CKD. If maintained longer-term, these should improve patient outcomes. Clinical trials of long-acting oral apelin agonists are justified in CKD and other conditions with impaired salt and water balance. Registration number at www.clinicalTrials.gov: NCT03956576. Funded by *Kidney Research UK*.

Chronic kidney disease (CKD) is common and is predicted to be the fifth leading cause of life-years lost by 2040[1]. Cardiovascular disease is independently associated with CKD and is its commonest complication[2]. As the estimated glomerular filtration rate (eGFR) declines, the risks of major adverse cardiovascular events and cardiovascular and all-cause mortality increase[2]. The current standard of care for CKD focuses on reducing blood pressure (BP) and proteinuria[3,4], ideally using renin-angiotensin system antagonists alongside the recent introduction of sodium-glucose co-transporter 2 (SGLT2) inhibitors. However, despite these, patient outcomes remain

e-mail: bean.dhaun@ed.ac.uk

poor, and newer treatments are urgently needed that will not only slow CKD progression but also reduce the associated risk of cardiovascular disease.

The apelin system comprises the apelin receptor and its two endogenous ligands, apelin and elabela[5]. Apelin is an endothelium-dependent vasodilator and a potent inotrope, and overall its actions oppose those of the renin-angiotensin system[6]. Clinical studies in healthy subjects and in patients with heart failure have shown that apelin lowers BP and systemic vascular resistance, and increases cardiac output[7,8]. With respect to the kidney, pre-clinical studies demonstrate that apelin regulates glomerular hemodynamics by opposing the actions of angiotensin II[9]. In addition, pre-clinical work shows that apelin opposes the actions of vasopressin[10], relevant for CKD as impaired salt and water balance is a feature and an important driver of hypertension. Thus, apelin offers major therapeutic promise in CKD. Here, we present the results of the first clinical studies examining the systemic and renal actions of apelin in health and CKD.

We find that the hemodynamic effects of apelin are equivalent in health and CKD: apelin lowers mean arterial pressure and systemic vascular resistance and increases cardiac output and renal blood flow. Apelin also promotes urinary sodium excretion and free water clearance. In patients with CKD alone, apelin reduces glomerular filtration rate, effective filtration fraction and proteinuria, and renoprotective effects analogous to an angiotensin-converting enzyme inhibitor. Overall, acute apelin treatment has beneficial cardiovascular and renal effects in patients with CKD who are receiving standard-of-care. Our data support studies of chronic apelin receptor agonism in this at-risk population.

## Results

In brief, we performed a randomised, double-blind, placebo-controlled, 2-way crossover study in 12 patients with stable CKD receiving recommended renoprotective treatment and in 12 age- and sex-matched healthy subjects. All subjects received pyroglutamate apelin-13 ([Pyr$^1$]apelin-13), 1 nmol/min and 30 nmol/min, or matched placebo on two separate study visits. Impedance cardiography assessed systemic hemodynamics, and infusions of para-aminohippurate and iohexol assessed renal blood flow and glomerular filtration rate, respectively.

Twenty-four subjects (12 with CKD and 12 healthy subjects) were recruited and completed both study visits with no adverse events. Both doses of [Pyr$^1$]apelin-13 were well tolerated. Data from all 24 subjects are included in the analyses. Baseline subject demographics are shown in Table 1. Compared to healthy subjects (48 ± 4 years, 67% male), patients with CKD (48 ± 4 years, 67% male) had a higher BP (systolic BP: 131 ± 3 *versus* 121 ± 3 mmHg) and increased arterial stiffness (pulse wave velocity: 6.5 ± 0.3 *versus* 5.7 ± 0.3 m/s). Patients with CKD had a reduced estimated glomerular filtration rate (35 ± 4 *versus* 100 ± 6 ml/min/1.73 m$^2$) and proteinuria (~ 0.5 g/day).

### Cardiovascular effects of [Pyr$^1$]apelin-13

Baseline cardiovascular parameters are shown in Table 2. As intended, only the infusion of 30 nmol/min [Pyr$^1$]apelin-13 increased plasma apelin concentration (Supplementary Fig. 1). Accordingly, infusion of [Pyr$^1$]apelin-13 1 nmol/min did not have demonstrable hemodynamic effects in either healthy subjects or patients with CKD (Supplementary Figs. 2–4).

### Apelin reduces BP and systemic vascular resistance in health and CKD.
[Pyr$^1$]apelin-13 30 nmol/min reduced mean arterial pressure (MAP) in both healthy subjects and those with CKD (health: 92 ± 2 to 89 ± 2 mmHg; CKD: 97 ± 2 to 94 ± 3; *p* < 0.05 for both), reductions of ~ 4% compared to placebo (Fig. 1A). Whereas systolic BP did not change in health, it fell from 131 ± 4 to 128 ± 3 mmHg in CKD (*p* < 0.05 compared to placebo, Supplementary Fig. 5). In healthy subjects, diastolic

BP fell from 78 ± 2 to 75 ± 2 mmHg and in CKD patients from 81 ± 2 to 77 ± 3 mmHg, (*p* < 0.05 for both compared to placebo, Supplementary Fig. 5), a reduction of ~ 5% in both groups.

Systemic vascular resistance fell during infusion of [Pyr$^1$]apelin-13 30 nmol/min (Fig. 1B). In healthy subjects, this fell by a maximum of 309 dynes*s*cm$^{-5}$m$^2$ compared to placebo (*p* < 0.01); in patients with CKD, this figure was 407 dynes*s*cm$^{-5}$m$^2$ (*p* < 0.01), reductions of 12% and 13%, respectively. Overall, 8 of 12 patients with CKD showed a > 10% reduction in systemic vascular resistance. There was no demonstrable between-group difference in response to apelin.

### Apelin increases cardiac index and heart rate in health and CKD.
In keeping with its inotropic actions, [Pyr$^1$]apelin-13 30 nmol/min increased cardiac index by ~ 10% in both healthy subjects and patients with CKD patients (2.7 ± 0.1 to 3.0 ± 0.1 L/min/m$^2$ and 2.5 ± 0.1 to 2.7 ± 0.2 L/min/m$^2$ respectively, both *p* < 0.05; Fig. 1C). Overall, patients with CKD showed a maximum increase in cardiac index of 15 ± 3% and three patients had a ≥ 20% rise. Whilst the effect was again short-lived in healthy subjects, it persisted to the end of the infusion in patients with CKD. Whereas apelin infusion was associated with an acute and short-lived 7% increase in heart rate in healthy subjects (56 ± 2 to 60 ± 2 bpm, *p* < 0.01 compared to baseline), heart rate did not change in patients with CKD (Supplementary Fig. 6).

### Apelin reduces arterial stiffness in health but not CKD.
In healthy subjects, pulse wave velocity fell by ~ 5% in response to [Pyr$^1$]apelin-13 30 nmol/min (6.3 ± 0.2 to 5.9 ± 0.2 m/s, *p* < 0.01 compared to placebo; Fig. 1D). This reduction in arterial stiffness was still developing when the apelin infusion had stopped and only returned to baseline after 45 min. Apelin did not affect arterial stiffness in patients with CKD.

### Renal effects of [Pyr$^1$]apelin-13

### Apelin increases renal blood flow in health and CKD.
Baseline renal parameters are shown in Table 2. In healthy subjects and in patients with CKD, both low and high doses of [Pyr$^1$]apelin-13 increased renal blood flow (Fig. 2A). In health, and compared to placebo, renal blood flow increased by ~ 140 mL/min and ~ 110 mL/min in response to [Pyr$^1$]apelin-13 1 nmol/min and 30 nmol/min, respectively (*p* < 0.01 for both), equivalent to increases of ~ 15%. In CKD, renal blood flow was still increasing at the end of both low and high-dose apelin infusions. Compared to placebo, and at maximum, [Pyr$^1$]apelin-13 1 nmol/min increased renal blood flow by ~ 40 mL/min and [Pyr$^1$]apelin-13 30 nmol/min by ~ 55 mL/min, increases of 8% and 20% (*p* < 0.05 and *p* < 0.01), respectively.

### Apelin reduces glomerular filtration rate in CKD.
Whereas infusion of [Pyr$^1$]apelin-13 did not affect glomerular filtration rate in healthy volunteers, both low and high dose [Pyr$^1$]apelin-13 reduced glomerular filtration rate (1 nmol/min: 41 ± 6 to 36 ± 6 mL/min; 30 nmol/min: 38 ± 5 to 34 ± 5 mL/min, *p* < 0.01 *versus* placebo for both) in patients with CKD (Fig. 2B). Notably, all patients with CKD experienced at least a 10% reduction in glomerular filtration rate in response to either dose of apelin.

### Apelin reduces filtration fraction and proteinuria in CKD.
In healthy subjects, [Pyr$^1$]apelin-13 1 nmol/min did not affect filtration fraction whereas [Pyr$^1$]apelin-13 30 nmol/min reduced filtration fraction from 19.3 ± 0.8 to 17.8 ± 0.7% (*p* < 0.05 *versus* placebo). In patients with CKD, and compared to placebo, both low and high doses of [Pyr$^1$]apelin-13 reduced filtration fraction (maximum response 1 nmol/min: 21.2 ± 1.2 to 19.4 ± 1.5%; 30 nmol/min: 22.0 ± 1.4 to 17.5 ± 0.8%; *p* < 0.0001 for both, Fig. 2C). As seen with renal blood flow, these effects were still developing at the end of the study period.

In keeping with the effects on filtration fraction, both [Pyr$^1$]apelin-13 1 nmol/min and 30 nmol/min reduced proteinuria in patients with

**Table 1 | Baseline characteristics of study subjects**

|  |  | Healthy subjects | CKD |
|---|---|---|---|
| n |  | 12 | 12 |
| Age, years |  | 48 ± 4 | 48 ± 4 |
| Male (%) |  | 8 (67) | 8 (67) |
| Smoking (%) | Current | 1 | 0 |
|  | Ex-smoker | 1 | 4 |
|  | Never | 10 | 8 |
| Clinical |  |  |  |
| Body mass index, kg/m$^2$ |  | 25 ± 1 | 28 ± 1 |
| Systolic blood pressure, mmHg |  | 121 ± 3 | 131 ± 3 |
| Diastolic blood pressure, mmHg |  | 75 ± 2 | 78 ± 2 |
| Mean arterial pressure, mmHg |  | 90 ± 2 | 95 ± 2 |
| Laboratory results |  |  |  |
| Haemoglobin, g/L |  | 133 ± 4 | 119 ± 5 |
| Na +, mmol/L |  | 139 ± 1 | 135 ± 1 |
| K +, mmol/L |  | 4.2 ± 0.1 | 4.5 ± 0.1 |
| Urea, mmol/L |  | 4.4 ± 0.4 | 13.2 ± 1.5 |
| Creatinine, mg/dL* |  | 0.85 ± 0.05 | 2.3 ± 0.23 |
| eGFR (ml/min/1.73 m$^2$) |  | 100 ± 6 | 35 ± 4 |
| CKD Stage 1 |  | – |  |
| 2 |  | – | 1 |
| 3 |  | – | 7 |
| 4 |  | – | 4 |
| 5 |  | – |  |
| Total cholesterol, mmol/L |  | 4.6 ± 0.2 | 4.5 ± 0.3 |
| LDL-cholesterol, mmol/L |  | 2.7 (2.0–2.8) | 2.4 (1.6–2.8) |
| Serum albumin, g/L |  | 36 ± 1 | 33 ± 1 |
| Urate, mmol/L |  | 0.31 ± 0.02 | 0.43 ± 0.02 |
| Urine protein:creatinine, mg/g |  | 0 | 411 (130–1193) |
| 24 hr urine Na +, mmol/24 hr |  | 98 ± 11 | 104 ± 11 |
| Primary Renal Disease |  |  |  |
| IgA nephropathy |  |  | 5 |
| ANCA-associated vasculitis |  |  | 2 |
| Hypertensive nephropathy |  |  | 2 |
| Anti-glomerular basement membrane disease |  |  | 1 |
| Reflux nephropathy |  |  | 1 |
| Chronic interstitial nephritis |  |  | 1 |
| Medications |  |  |  |
| Angiotensin-converting enzyme inhibitor (%) |  |  | 7 (58) |
| Alpha-blocker (%) |  |  | 2 (17) |
| Angiotensin receptor blocker (%) |  |  | 3 (25) |
| Beta-blocker (%) |  |  | 3 (25) |
| Calcium channel blocker (%) |  |  | 4 (33) |
| Diuretic (%) |  |  | 3 (25) |
| Immunosuppression (%) |  |  | 3 (25) |
| Sodium-glucose co-transporter 2 inhibitor (%) |  |  | 2 (17) |
| Statin (%) |  |  | 4 (33) |

ANCA anti-neutrophil cytoplasm antibody, IgA immunoglobulin-A, LDL low-density lipoprotein. Data are mean±SEM or median (interquartile range). *To convert to µmol/L multiply by 88.4.

CKD (1 nmol/min: 349 ± 98 to 269 ± 91 µg/min; 30 nmol/min: 308 ± 86 to 252 ± 73 µg/min, $p < 0.01$ for both, Fig. 2D), equivalent to reductions of ~25%. There was no relationship between baseline glomerular filtration rate and the degree of proteinuria reduction.

**Table 2 | Baseline hemodynamic and renal measures**

|  | Healthy subjects | CKD |
|---|---|---|
| n | 12 | 12 |
| Heart rate, bpm | 58 ± 2 | 59 ± 2 |
| Cardiac index, L/min/m$^2$ | 2.9 ± 0.1 | 2.6 ± 0.2 |
| Systemic vascular resistance index, dynes*sec*cm$^{-5}$ m$^2$ | 2,429 ± 87 | 2,862 ± 180 |
| Pulse wave velocity, m/s | 5.7 ± 0.3 | 6.5 ± 0.3 |
| Central augmentation index, % | 3.6 ± 3.9 | 7.5 ± 4.1 |
| Effective renal blood flow, mL/min | 824 ± 48 | 313 ± 49 |
| Effective renal vascular resistance, mmHg/min/L | 117 ± 8 | 405 ± 73 |
| Effective filtration fraction, % | 19.5 ± 0.9 | 20.7 ± 1.2 |
| Glomerular filtration rate, mL/min | 96 ± 5 | 41 ± 5 |
| Urine sodium excretion, µmol/min | 217 ± 18 | 233 ± 23 |
| Protein excretion rate, µg/min | 14 (5–127) | 192 (70–525) |
| Free water clearance, mL/min | 5.7 ± 0.4 | 3.9 ± 0.5 |

The data shown are an average of baseline measurements from each study visit for each participant. Augmentation index has been corrected to 75 bpm. Data are mean±SEM or median (interquartile range).

**Apelin promotes salt and water excretion in health and CKD**. Both doses of [Pyr$^1$]apelin-13 increased urinary sodium excretion in healthy subjects and patients with CKD (Fig. 3A). In health, low dose [Pyr$^1$] apelin-13 increased natriuresis by ~40 µmol/min in comparison to placebo ($p < 0.01$), a rise of ~30%. Most healthy subjects had a ≥20% increase in natriuresis at this dose. The effects of high-dose [Pyr$^1$] apelin-13 were similar. In CKD, and compared to placebo, [Pyr$^1$]apelin-13 1 nmol/min increased urinary sodium excretion by ~40 µmol/min ($p < 0.0001$), with a similar response seen at [Pyr$^1$]apelin-13 30 nmol/min. These increases in natriuresis were still developing at the study end. Overall, apelin increased urinary sodium excretion in patients with CKD by ~30%.

Finally, we examined the effects of apelin on the clearance of free water. In healthy subjects, [Pyr$^1$]apelin-13 1 nmol/min increased free water clearance by 1.2 ± 0.5 mL/min compared to placebo ($p < 0.05$), an increase of ~25%. The higher dose of apelin had no effect. In patients with CKD, both low and high doses of apelin increased free water clearance. [Pyr$^1$]apelin-13 1 nmol/min increased free water clearance from 3.8 ± 0.4 mL/min to a maximum of 4.1 ± 0.4 mL/min ($p < 0.05$). There was a similar magnitude of response to [Pyr$^1$]apelin-13 30 nmol/min. If maintained over a 24-hour period, this would equate to an increase in free water clearance of ~0.5 L.

## Discussion

Chronic kidney disease (CKD) is an increasingly common public health concern. With a current global prevalence of ~10%, it now ranks as the twelfth leading cause of death worldwide[11]. Importantly, patients with CKD are more likely to die from cardiovascular diseases than they are to reach kidney failure and the need for dialysis or kidney transplant[2,12]. Indeed, almost 8% of global cardiovascular deaths in 2017 were attributable to CKD[11]. Unfortunately, these risks continue despite the current standard of care, and new add-on treatments are urgently needed. The ideal treatment would provide direct renoprotection, whilst also offering broad cardiovascular protection. In this context, our study had four main findings. First, we demonstrated that apelin lowers BP and causes peripheral vasodilation in patients with CKD. Second, apelin increases renal blood flow and reduces renal vascular resistance. Third, we observed important reductions in filtration fraction and proteinuria in patients with CKD. Fourth, apelin promotes salt and water excretion. Our experimental design allowed us to show that these cardioprotective and renoprotective effects are observed on top

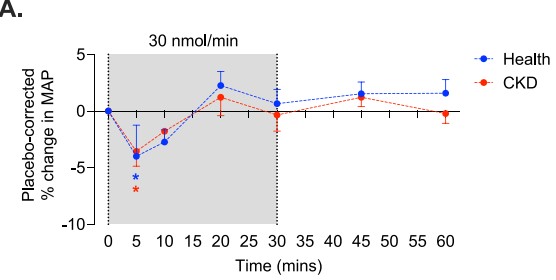

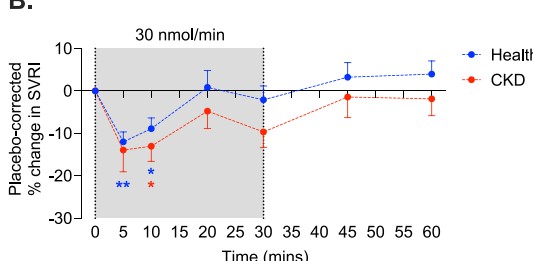

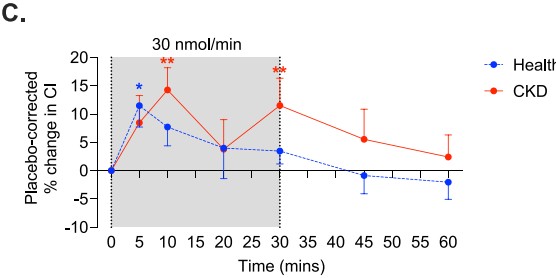

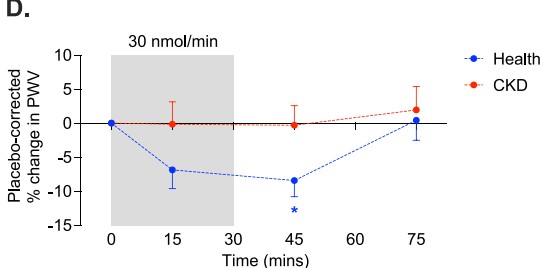

**Fig. 1 | Apelin improves systemic haemodynamics in health and chronic kidney disease.** In healthy subjects (blue, *n* = 12) and patients with chronic kidney disease (CKD, red, *n* = 12), 30 nmol/min [Pyr¹]apelin-13 reduced mean arterial pressure (MAP, **A**) in health (*\*p* = 0.045) and CKD (*\*p* = 0.027) for within-group comparisons to baseline. Systemic vascular resistance index (SVRI, **B**) was also reduced by 30 nmol/min [Pyr¹]apelin-13 (*\*p* = 0.025, *\*\*p* = 0.004 in health and *\*p* = 0.018 in CKD for within-group comparison to baseline). 30 nmol/min [Pyr¹]apelin-13 increased

cardiac index (CI, **C**) in health (*\*p* = 0.043) and CKD (*\*\*p* < 0.01) compared to baseline. In healthy subjects only, 30 nmol/min [Pyr¹]apelin-13 reduced pulse wave velocity (**D**) (*\*p* = 0.011) compared to baseline. The data shown are placebo-corrected responses. The grey bar represents the time during which apelin was infused. Data are presented as mean values +/− SEM. Analyses by mixed-effects model (MAP, SVRI and CI) or 2-way ANOVA (PWV) with Dunnett's multiple comparison correction. Source data are provided as a Source Data file.

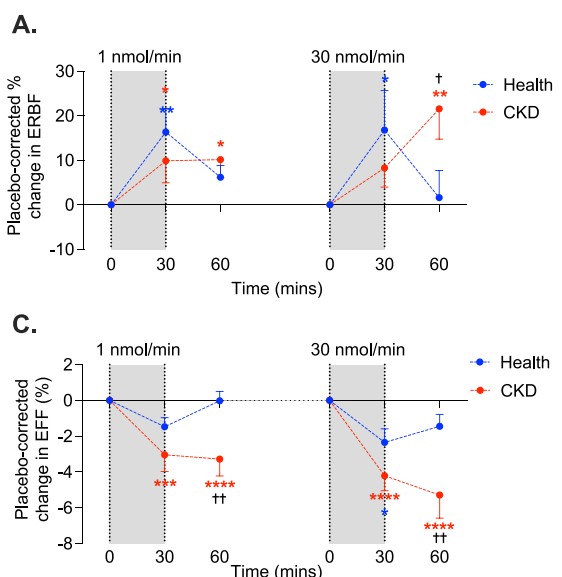

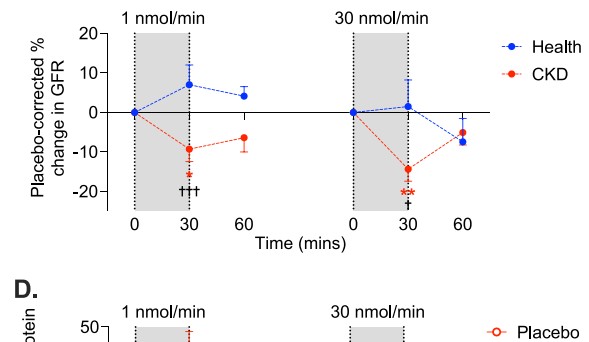

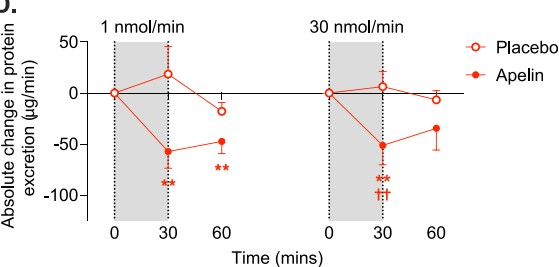

**Fig. 2 | Apelin alters renal haemodynamics in health and chronic kidney disease.** Both 1 nmol/min and 30 nmol/min infusions of [Pyr¹]apelin-13 altered renal haemodynamics in healthy subjects (blue, n = 12) and in patients with chronic kidney disease (CKD, red, n = 12). Effective renal blood flow (ERBF) increased in both groups (**A**) (*\*p* = 0.044, *\*\*p* = 0.002 in health and *\*p* < 0.05, *\*\*p* = 0.006 in CKD for within-group comparison to baseline; ⁺*p* = 0.038 for between-group comparison). Glomerular filtration rate (GFR) fell in patients with CKD alone (**B**) (*\*p* = 0.028, *\*\*p* = 0.001 for within-group comparison to baseline and ⁺*p* = 0.015, ⁺⁺⁺*p* = 0.0009 for between-group comparison). Effective filtration fraction (EFF) fell in both health

and CKD (**C**) (*\*p* = 0.045 in health and *\*\*\*p* = 0.0002, *\*\*\*\*p* < 0.0001 in CKD for within-group comparison to baseline and ⁺⁺*p* < 0.01 for between-group comparison). In patients with CKD, urinary protein excretion fell significantly (**D**) (*\*\*p* < 0.01 for within-group comparison to baseline and ⁺⁺*p* = 0.006 for comparison to placebo). The grey bar represents the time during which apelin was infused. Data are presented as mean values +/− SEM. Analyses were by 2-way ANOVA (GFR) or mixed effects model (ERBF, EFF and protein excretion) with Dunnett's or Šidák's multiple comparison corrections for within or between-group comparisons, respectively. Source data are provided as a Source Data file.

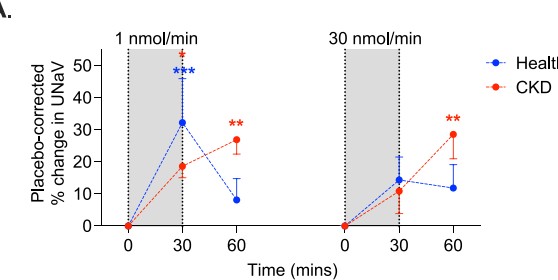

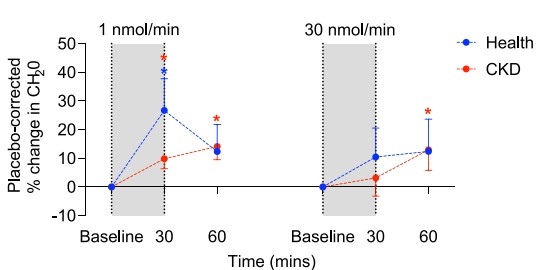

**Fig. 3 | Apelin promotes natriuresis and free water clearance in health and chronic kidney disease.** In both healthy subjects (blue, $n = 12$) and patients with chronic kidney disease (CKD, red, $n = 12$) 1 nmol/min and 30 nmol/min [Pyr$^1$]apelin-13 increased sodium excretion (UNaV, **A**) (***$p = 0.0005$ in health and *$p = 0.033$,**$p < 0.01$ in CKD for within-group comparison to baseline). Both doses of apelin also increased free water clearance (CH$_2$0, **B**) (*$p = 0.037$ in health and *$p < 0.05$ in CKD for within-group comparison to baseline). The grey bar represents the time during which apelin was infused. Data are presented as mean values +/− SEM. Analyses were by mixed effects model with Dunnett's multiple comparison correction. Source data are provided as a Source Data file.

of renin-angiotensin system blockade and that apelin has direct renal effects. Overall, apelin was safe and well tolerated in CKD patients and our data define the translational potential of apelin as a new therapy for CKD and its associated cardiovascular disease.

These are the first clinical studies of apelin in kidney disease. In keeping with the favourable systemic hemodynamic effects in heart failure[7,8], we demonstrate that apelin reduces BP, causes peripheral vasodilation, and increases cardiac output in patients with CKD. Hypertension is a frequent finding in patients with CKD[13] and despite treatment with multiple antihypertensive agents, most patients with CKD fail to reach target BP[14]. Although the BP-lowering effect of apelin was modest (~ 4 mmHg), it is similar in magnitude to that of the SGLT2 inhibitor, empagliflozin, which lowers systolic BP by ~ 3 mmHg and diastolic BP by ~ 1 mmHg in CKD, and reduces cardiovascular and all-cause mortality in those with and without CKD[15,16]. More broadly, a 5 mmHg reduction in systolic BP is recognised to reduce the risk of major cardiovascular events by ~ 10%[17]. The effect of apelin on BP might have been more impressive had the subjects not had such good baseline BP control.

Epidemiological data demonstrate that arterial stiffness, measured by pulse wave velocity, is an independent risk factor for cardiovascular morbidity and mortality[18,19]. In the current study, apelin reduced pulse wave velocity by ~ 5% in healthy subjects compared to placebo. The magnitude of this reduction is similar to that of a renin-angiotensin system inhibitor[20,21], and the effect is likely due to the reduction in BP seen with apelin. However, we did not observe vascular unstiffening in patients with CKD despite a comparable fall in BP. It is possible that the structural and functional vascular changes characterising CKD (e.g., medial vascular calcification) are less likely to be overcome acutely and a longer duration of treatment is needed. Notably, in pre-clinical models of pulmonary arterial hypertension, apelin agonism improves vascular hemodynamics and vascular

remodelling[22,23]. If chronic apelin receptor agonism could reduce arterial stiffness in CKD, it would be expected to improve patient outcomes more than BP reduction alone[24].

Changes in systemic hemodynamics will affect renal hemodynamics. Our protocol was designed to examine both indirect and potential direct effects of apelin on the kidney. The higher apelin dose (30 nmol/min) was expected to affect systemic (and therefore renal) hemodynamics, and the lower sub-systemic dose (1 nmol/min) was used to define direct renal effects. Whereas only the higher dose of apelin affected systemic hemodynamics, both low and high doses of apelin affected renal hemodynamics supporting a direct effect of apelin on the kidney.

Both low and high-dose apelin increased renal blood flow in healthy subjects and subjects with CKD. These findings have two implications. First, that apelin causes glomerular afferent arteriolar vasodilation and second, that a downregulated apelin system contributes to the increased renovascular tone seen in CKD. An increase in renal blood flow might be expected to increase the glomerular filtration rate. However, the glomerular filtration rate did not change in healthy subjects, suggesting that apelin has a similar vasodilatory effect at the efferent arteriole. In patients with CKD, we observed a fall in glomerular filtration rate which suggests that apelin preferentially vasodilates the efferent arteriole, although not excluding an effect on mesangial cells and filtration coefficient. Our data are consistent with pre-clinical studies[9], and these hemodynamic effects of apelin are similar to those of an endothelin blocker[25], agents that have recently been licensed for CKD[26].

In line with the effects of apelin on efferent arteriolar tone and glomerular perfusion pressure, we observed a reduction in filtration fraction and proteinuria in those with CKD. These effects are analogous to but occur in addition to, those seen with renin-angiotensin system blockade[27]. Proteinuria reduction is important both for reducing the risk of CKD progression[28] and its associated cardiovascular disease[29,30]. However, despite the standard of care, many patients with proteinuric CKD have residual proteinuria[31]. In the current study, all subjects were established on maximally tolerated treatment with an angiotensin-converting enzyme inhibitor or angiotensin receptor blockers with good BP control. Despite this, mean baseline proteinuria remained at ~ 0.5 g/d. Interestingly, the magnitude of proteinuria reduction we see (~ 25%) is similar to that observed with an endothelin receptor antagonist and an SGLT2 inhibitor[25,32].

In both healthy subjects and those with CKD, apelin promoted natriuresis. If the natriuretic response observed was maintained over 24 h, then patients would excrete an extra ~ 60 mmol of sodium, which is equivalent to the effect of 25 mg of spironolactone[33]. Natriuresis was seen with both doses of [Pyr$^1$]apelin-13 but was more impressive with the lower dose. This might be expected given the higher dose of apelin-affected systemic hemodynamics in a manner that should promote salt (and water) retention by the kidney. We have previously shown that apelin and its receptor are expressed throughout the human kidney[34], including nephron segments intimately involved in regulating sodium balance, so direct tubular action is not unexpected. In addition, pre-clinical studies have shown that apelin decreases epithelial sodium channel expression and activity[35], which would lead to reduced sodium reabsorption, although we recognise that the effects seen here are probably too rapid to be explained by this. Finally, we have previously shown co-localisation of the apelin receptor and the SGLT2 in the proximal tubule[34]. SGLT2 inhibitors are also recognised to promote natriuresis[36], and a synergistic crosstalk between apelin and SGLT2 to promote salt excretion is an intriguing possibility that should be explored in future work.

In keeping with pre-clinical studies[9,37], apelin increased the clearance of free water – the volume of solute-free water excreted by the kidneys. The mechanism for this may relate to an inhibition of vasopressin signalling. Apelin and vasopressin show reciprocal regulation

in humans in response to changes in osmolality, and in vitro and in vivo studies find apelin directly inhibits the actions of vasopressin in the collecting duct, thus promoting free water loss[37–39]. Notably, vasopressin also activates the epithelial sodium channel, promoting distal sodium reabsorption, and so its downregulation by apelin may be another mechanism whereby apelin promotes natriuresis[40]. Overall, our findings are relevant to conditions beyond CKD where salt and water excretion is impaired, such as heart failure, liver disease and syndrome of inappropriate antidiuresis.

The effects of apelin on renal parameters appeared delayed in patients with CKD compared to healthy subjects, with a suggestion of prolonged and perhaps increased systemic hemodynamic effects in this group. There are several potential explanations for these differences. The apelin system is expressed throughout the human kidney[34], and whilst there is no published human data, the system is downregulated in animal models of CKD[41]. Thus, we expect the system to be modified in patients with CKD. The relationship between the apelin system and the renin-angiotensin system is also implicated. There is a crosstalk between the systems. For example, in a model of heart failure, angiotensin II-induced downregulation of the apelin receptor is restored by angiotensin receptor blockade[42]. In our study, most patients with CKD were established on renin-angiotensin system blockers, and these drugs may alter apelin signalling. Inhibition of angiotensin II may also affect the speed at which compensatory mechanisms are able to respond to acute changes in systemic hemodynamics.

We acknowledge the limitations of our data. This is an acute study, and it will be important to confirm that these effects are maintained longer-term. In addition, we studied a relatively homogeneous CKD population, and further work is needed in a broader population of patients with CKD, including those with diabetes, renal vascular disease, and active multi-system inflammatory disease. However, our patients were at elevated cardiovascular risk as reflected by their increased arterial stiffness and proteinuria. Our study was also only powered to examine the effects of apelin on systemic vascular resistance and renal blood flow. Finally, although optimised on antagonists of the renin-angiotensin system, only a minority of subjects were receiving SGLT2 inhibitors as these were not recommended for use in CKD until the final months of our study. Future work in this area should examine the role of apelin agonism in patients with CKD receiving both renin-angiotensin system and SGLT2 inhibitors.

In conclusion, we have shown in a randomised, double-blind, placebo-controlled crossover study that apelin has beneficial systemic and renal effects in healthy subjects, and these effects are preserved and accentuated in patients with CKD already optimised on current treatment. If these effects were maintained longer-term, they would be expected to reduce cardiovascular and renal risk and improve patient outcomes. Our data, therefore, provide justification for future studies in kidney disease and beyond. Encouragingly, this will be facilitated by long-acting, orally available apelin receptor agonists, which are in development[43].

## Methods

### Study design and oversight
We performed a randomised, double-blind, placebo-controlled crossover study in patients with chronic kidney disease and age- and sex-matched healthy subjects. Recruitment and all studies were completed at the University of Edinburgh between May 2021 and December 2022 in accordance with the principles of the Declaration of Helsinki. The study ended when the recruitment target was met. Ethical approval was obtained from the South-East Scotland Research Ethics Committee (Reference18/SS/0145). All subjects provided written informed consent prior to participation, and the study was registered at www.clinicaltrials.gov (NCT 03956576). The study protocol is provided.

### Subjects
Adult patients with stable CKD stages 1–4 were recruited from the renal outpatient clinic within NHS Lothian. To be included, subjects had to be over the age of 18 years and able to give informed consent. Stable CKD was defined as < 10% variability in eGFR over at least two measurements taken three months apart. Where the primary renal disease was due to a multi-system inflammatory condition (e.g., ANCA-associated vasculitis), this was in clinical remission. Exclusion criteria were: overt cardiovascular disease or a diagnosis of diabetes mellitus as these are recognised to affect the apelin system[44,45]; kidney failure (estimated glomerular filtration rate < 15 mL/min/1.73 m$^2$) or receiving kidney replacement therapy (including kidney transplant recipients); serum albumin < 30 g/L; a diagnosis of polycystic kidney disease or tolvaptan therapy due to the relationship between apelin and vasopressin[10]; multiple or severe drug or food allergies; pregnancy or breastfeeding; uncontrolled hypertension (BP > 160/100 mmHg). Sex was self-reported. Age- and sex-matched healthy subjects were recruited from local ethically approved databases. 'Health' was defined as no current or previous medical issues and no regularly prescribed medication.

### Randomisation
A randomised and numbered study visit schedule was generated prior to subject recruitment. This specified the order in which subjects received apelin or placebo. Upon recruitment by the investigator, the study nurse randomly allocated each subject to a numbered visit schedule to which both the investigator and subject were blinded. The investigators remained blinded to the visit schedule until all study visits were complete.

### Drugs
Pyroglutamate apelin-13 ([Pyr$^1$]apelin-13, Severn Biotech Ltd., Kidderminster, UK) was dissolved in 5 ml 0.9% saline and administered at 1 and 30 nmol/min for 30 min as two separate infusions with a 90-min washout between doses (Supplementary Fig. 7). These doses were based on dose-ranging studies and our previous work[7,8]. In these studies, 1 nmol/min [Pyr$^1$]apelin-13 caused local vasodilation but had no discernible effects on systemic hemodynamics. Therefore, we hypothesised that it might allow us to identify isolated renal hemodynamic effects of apelin. The higher 30 nmol/min dose had significant systemic hemodynamic effects in both local and systemic vascular studies.

Iohexol (Omnipaque$^{TM}$ 140 mg Iodine/mL, 200 ml, GE Healthcare, Chicago, USA) and para-aminohippurate (PAH, Sodium 4-aminohippurate 2 g in 10 ml, Clinalfa AG and Basic Pharma, Geleen, The Netherlands) were diluted in 100 ml 5% dextrose and a weight-based loading dose was given over 15 min, followed by a maintenance infusion. For subjects with an estimated glomerular filtration rate < 60 mL/min/1.73 m$^2$, doses of PAH and iohexol were reduced as in our previous studies[25].

### Assays
At prespecified time points (Supplementary Fig. 7), venous blood was collected into EDTA tubes (Monovette® Sarstedt, Numbrecht, Germany) for the measurement of plasma apelin, haematocrit, iohexol, and PAH, and serum gel tubes (Monovette® Sarstedt, Numbrecht, Germany) for measurement of serum sodium, potassium, and osmolality. In addition, 20 ml aliquots of urine from each void were collected into plain universal containers for the measurement of PAH, iohexol, sodium, potassium, and protein.

Haematocrit was measured immediately on whole blood using a Coulter counter in the hospital laboratory. All other blood samples were centrifuged within 90 min of sampling (2500 × g for 20 min for EDTA, 3000 × g for 15 min for gel) with plasma, serum and urine samples, then stored at − 80 °C until required. [Pyr$^1$]apelin-13

concentration was measured using an enzyme-linked immunosorbent assay (EK-057-23, Phoenix Pharmaceuticals, Burlingame, USA). Plasma and urine iohexol and PAH concentrations were measured using high-performance liquid chromatography (HPLC) with ultraviolet (UV) detection[46,47]. Plasma and urine sodium and potassium concentrations were measured using an ion-selective electrode (Roche Diagnostics AVL 9180 series electrolyte analysers, Fisher Scientific, Leicestershire, UK)[48] and osmolality using an osmometer (Loser micro-digital osmometer, Camlab, Cambridge, UK). Urine protein was measured using a Bradford assay[49] with volumes adapted for use on a Cobas Fara or Mira analyser (Roche Diagnostics Ltd, Welwyn Garden City, UK).

### Cardiovascular assessments

BP was measured using the Omron HEM-705CP, a validated oscillometric sphygmomanometer[50]. BP was recorded as an average of two readings from the right arm with $\leq 10$ mmHg difference. Mean arterial pressure was calculated (diastolic BP + 1/3 pulse pressure). Arterial stiffness was measured by gold standard pulse wave velocity using the SphygmoCor® system (SphygmoCor® Mx, AtCor Medical, Sydney, Australia, version 6.31)[51]. A high-fidelity micromanometer (SPC-301, Millar Instruments, Texas, USA) was used to determine carotid-femoral pulse wave velocity. Values were recorded as the average of two measurements within 0.5 m/s. Thoracic electrical impedance was performed using a validated impedance cardiograph, the BioZ® Dx (CardioDynamics®, Sonosite, Bedford, UK), with the non-invasive assessment of heart rate (beats per min), cardiac output (L/min) and systemic vascular resistance (dynes*sec*cm$^{-5}$). Data were corrected for body surface area to give cardiac index (L/min/m$^2$) and systemic vascular resistance index (dynes*sec*cm$^{-5}$m$^{-2}$), respectively, to allow direct comparison between subjects. Measurements were recorded as the average of two consecutive readings with < 10% difference between the two. If the required criteria for measurements were not met, readings were repeated until they were achieved, or the data point was omitted at the discretion of the investigator in the overall interests of the study.

### Study protocol

Studies were performed in the University of Edinburgh's Clinical Research Centre, Western General Hospital. All studies were carried out in a quiet, temperature-controlled room (22–24 °C). Subjects were asked to adhere to a low-salt diet for 3 days prior to each study day and to abstain from smoking, alcohol, and caffeine for 24 hours prior to each study. Subjects completed a 24-hour urine collection the day prior to each study. All subjects attended for two study days separated by at least 2 weeks. They received systemic infusions of [Pyr$^1$]apelin-13 on one visit and matched placebo on the other.

Subjects received a light breakfast on arrival and thereafter remained supine throughout the study except when voiding. Routine medications were taken other than diuretics and SGLT2 inhibitors (omitted for 24 h and 7 days, respectively, prior to each study). Three cannulae (two 18 G and one 20 G) were inserted into the forearms. Diuresis was induced by a bolus of 500 ml 5% dextrose given over 30 min, with a loading dose of iohexol and PAH commenced 15 min later. Thereafter, maintenance infusions of iohexol and PAH and 5% dextrose to a total volume of 400 mL/h continued throughout the study. A 2.5-h equilibration period was followed by either infusion of [Pyr$^1$]apelin-13 or placebo. Cardiovascular assessments and blood sampling were performed at pre-specified time points. Urine was collected by spontaneous voiding every 30 min (Supplementary Fig. 7, created with BioRender.com, released under a Creative Commons Attribution-NonCommercial-NoDerivs 4.0 International license).

### Study endpoints and data analysis

Pre-specified coprimary endpoints were changes in systemic vascular resistant index and renal blood flow. Secondary endpoints were changes in cardiac output, blood pressure, pulse wave velocity, glomerular filtration rate, natriuresis, free water clearance and urinary protein excretion.

Glomerular filtration rate and effective renal plasma flow were calculated from iohexol and PAH clearances, respectively, as follows:

Glomerular filtration rate = Urine$_{Iohexol}$ / Plasma$_{Iohexol}$ x urine flow rate (mL/min)

Effective renal plasma flow = Urine$_{PAH}$ / Plasma$_{PAH}$ x urine flow rate (mL/min)

Effective renal blood flow was calculated by dividing the effective renal plasma flow by (1-haematocrit) and effective renal vascular resistance by dividing mean arterial pressure by effective renal blood flow. Urinary protein and sodium excretion were calculated as their respective urinary concentrations multiplied by urinary flow rate. Effective filtration fraction was calculated as the ratio of glomerular filtration rate to effective renal plasma flow and expressed as a percentage.

### Sample sizes & statistics

As apelin has not been given to patients with CKD before, sample sizes were based on our previous studies in healthy subjects and patients with heart failure. For the coprimary endpoints of systemic vascular resistance index and renal blood flow, a sample size of 12 subjects per group would have 85% power to detect a 10% reduction in systemic vascular resistance index and a 15% increase in renal blood flow in response to apelin compared to placebo with a two-sided 5% significance level[7,8]. Previous studies examining apelin in heart failure have included similar sample sizes[7,8].

All data were tested for normality with log transformation as appropriate. For the purpose of analyses, sex/gender were grouped together due to the small sample size. Data are presented as mean ± standard error of the mean (SEM) or median (interquartile interval) as appropriate. Data were analysed using paired and unpaired $t$ tests or non-parametric equivalents as appropriate, analysis of variance (ANOVA) with repeated measures and multiple comparison corrections or mixed effect analysis. For ANOVA and mixed effect models, where the model identified a significant result, the change in the variable of interest was then compared at prespecified time points to baseline (within-group comparison, using Dunnett's multiple comparison correction) and between groups (using Šidák's multiple comparison correction). Simple linear regression was used to explore relationships between variables, with Pearson correlation coefficients. Results were considered statistically significant if a two-sided $p$-value was < 0.05. All statistical analyses were carried out on Prism® (version 9.4.1, GraphPad Software Inc, San Diego, USA).

### Reporting summary

Further information on research design is available in the Nature Portfolio Reporting Summary linked to this article.

## Data availability

Source data are provided in this paper. De-identified individual participant data are available on request from the corresponding author (bean.dhaun@ed.ac.uk) for up to 6 months from the online publication date without restrictions. A full study protocol is available on request. Source data are provided in this paper.

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

## Acknowledgements

F.A.C. was funded through a Clinical Research Training Fellowship from Kidney Research UK (TF_006_20171124) with additional grants from the Scottish Society of Physicians and Mason Medical Research Trust. A.P.D. and J.J.M. received funding from the Wellcome Trust (203151/Z/16/Z). D.E.N. is funded by the British Heart Foundation (CH/09/002/26360, RE/18/5/34216, RG/F/22/110093). N.D. is supported by a Senior Clinical Research Fellowship from the Chief Scientist Office (SCAF/19/02).

## Author contributions

F.A.C., J.J.M., A.P.D., D.E.N., and N.D. designed the study; F.A.C. recruited all participants; F.A.C., V.M., E.G. and B.M. completed all study visits; L.B. performed laboratory analyses; F.A.C. performed statistical analyses; F.A.C. and N.D. wrote the manuscript; all authors critically appraised and approved the final manuscript.

## Competing interests

D.E.N. has historically received consultancy fees from Bristol-Myers Squibb and Amgen regarding the development of apelin receptor agonists for heart failure and was a sub-investigator in a clinical trial for Bristol-Myers Squibb (NCT03281122). All other authors have no competing interests.

## Additional information

Fiona A. Chapman[1,2], Vanessa Melville[1], Emily Godden[1], Beth Morrison[1], Lorraine Bruce[1], Janet J. Maguire ◉ [3], Anthony P. Davenport ◉ [3], David E. Newby ◉ [1] & Neeraj Dhaun ◉ [1,2] ✉

[1]University/BHF Centre for Cardiovascular Science, The Queen's Medical Research Institute, University of Edinburgh, Edinburgh, UK. [2]Department of Renal Medicine, Royal Infirmary of Edinburgh, Edinburgh, UK. [3]Division of Experimental Medicine and Immunotherapeutics, Addenbrooke's Centre for Clinical Investigation, University of Cambridge, Cambridge, UK. ✉e-mail: bean.dhaun@ed.ac.uk

