## [Peer Review File · Nature Communications]

Cardiovascular and renal effects of apelin in chronic kidney disease: a randomized, double-blind, placebo-controlled, crossover studyEditorial note: Parts of this Peer Review File have been redacted as indicated to remove third-party material where no permission to publish could be obtained.

REVIEWER COMMENTS

Reviewer #1 (Remarks to the Author):

I was pleased to be asked by the Editors of Nature Communications to review the paper "Cardiovascular and renal effects of apelin in chronic kidney disease: A randomised, double-blind, placebo-controlled, crossover study"

- What are the noteworthy results?

Apelin exerts effects on blood pressure, systemic vascular resistance, increased cardiac index, increased renal blood flow and increased natriuresis and free water clearance; all of which has not been reported in such detail before.

- Will the work be of significance to the field and related fields? How does it compare to the established literature? If the work is not original, please provide relevant references.

This work is novel and no prior studies in the field have systematically examined effects of apelin on experimental endpoints as used by the authors.

- Does the work support the conclusions and claims, or is additional evidence needed?

The work presented herein supports the conclusions by the authors.

- Are there any flaws in the data analysis, interpretation and conclusions? - Do these prohibit publication or require revision?

No.

- Is the methodology sound? Does the work meet the expected standards in your field?

Yes.

- Is there enough detail provided in the methods for the work to be reproduced?

Yes.

Specific comments:

The effect of apelin on systolic and diastolic blood pressure (supplementary figure 4) seems to vanish at 20 minutes and there seems to be a compensatory increase thereafter. Do the authors have any explanations for this phenomenon?

Data on heart rate would be interesting and authors might provide these in a supplemental file. How do the authors explain differences observed between healthy individuals and CKD patients? Does it correlate with arterial stiffness?

Another interesting finding is that alterations to the blood flow seem to occur at a later stage in the CKD population. How do authors explain this? Does that eventually mean that full effects of apelin are somehow delayed in the CKD population?

It is of course speculative, but what kind of direct effects would the authors expect when discussing renal haemodynamics?

Not all patients were on RASi and as mentioned by the authors, only a minority on SGLT2i. Was there a difference in proteinuria reduction when no RASi was used or in those where SGLT2i were introduced?

One typo: (p8, line 192) should be our, otherwise very well written.

I slightly disagree when authors mention a homogeneous population and then mention vasculitis - overall, 3 patients had a vasculitis.

Did you see a trend for more pronounced effects when apelin was used in more severe CKD stages, e.g. in those with a GFR of below 30?

Methods: how was stable CKD defined?

Assays: the authors mention that apelin concentrations were measured by an ELISA, but such results are not given. The same is true for iohexol and PAH concentrations.

Importantly, did any of the patients report any side effects when the low dose or the higher dose of apelin was administered?

Reviewer #2 (Remarks to the Author):

This manuscript presents a meticulously conducted clinical study focusing on a novel approach to treating CKD, showcasing exemplary methodological rigor. Despite its modest scale as a proof-of-concept investigation, it unveils compelling therapeutic prospects. The authors are recognized experts in the field with multiple publications on the subject.

Our questions / comments to the authors are the following

1) In your previous work (Reference 34), you demonstrated higher apelin levels in women. Did you observe any sex differences in the effects of apelin within your study population?

2) In this same previous work (Reference 34), you demonstrated an inverse relationship between apelin levels and GFR. Looking at this finding, it seems reasonable to hypothesize a certain level of apelin resistance in patients with CKD. However, you show in this new study that certain effects are either more pronounced or prolonged in CKD, which is even more surprising knowing that apelin receptors are prone to desensitization and internalization. What are the authors thoughts on this matter ? Could the pharmaceutical RAAS inhibition play a role here?

3) In this same previous work (Reference 34), you demonstrated a colocalization of apelin, apelin receptor and SGLT2 in the kidneys, and you now show that apelin and SGLT2 inhibitors share certain effects, e.g., on proteinuria.

a. Do you think that there might be a crosstalk between both pathways ?

b. The natriuresis induced by SGLT2 inhibitors has been shown to be transient (Rao et al., JASN, 2023). Do you think that the natriuresis induced by apelin might also be transient ?

4) Concerning the two different doses of apelin, you refer to two publications with dose-response experiments. Some more information in the manuscript would be helpful for a better understanding of the choice of these two different doses.

Also, some comment would be useful about what dose would be most suitable in the opinion of the authors for further evaluation in CKD?

5) Can you hypothesize why an increase in heart rate is observed in healthy subjects but not in CKD patients?

6) You discussed the interplay between apelin and angiotensin II and arginine vasopressin (AVP). You excluded patients with ADPKD treated with tolvaptan to avoid interference with AVP activity. However, most of your CKD patients were treated with either an angiotensin-converting enzyme inhibitor or an angiotensin receptor blocker. Do you believe this could influence the impact of apelin on your study outcomes?

7) You may want to consider discussing certain limitations in implementing apelin analogs for treating CKD.

a. The decrease in blood pressure appears to be primarily driven by the reduction in diastolic blood pressure. Why do you think this is the case, and do you believe it could limit its use in patients with coronary heart disease, considering that coronary blood flow greatly depends on diastolic pressure?

b. As you mentioned, apelin is rather thought to act in a paracrine or autocrine manner. Systemic administration of long-acting apelin analogs inevitably raises concerns about selectivity. Of particular concern is the angiogenic effect of apelin, especially in oncology patients and those with diabetes mellitus and diabetic retinopathy.

8) In your introduction, you mentioned another apelin receptor ligand « elabela ». Would you anticipate that elabela might have a different effect profile compared to apelin?

Reviewer #4 (Remarks to the Author):

Sound research and very clear write up. Promising results which I hope will be confirmed in future phase II/III studies. I comment as a statistician with no expertise on the medical background/discussion.

The study appears to be well-powered. But why was a power of 85% chosen? Given the relatively small sample size you could have aimed for 90% power. What was your rationale? Please explain.

I like the figures/visualisations, however as a statistician I can't recommend publication without a more detailed reporting of the outcomes of the ANOVA and mixed models. I want to see 95% confidence intervals and SE.

You are only powered for null-hypothesis testing for the primary endpoint. Thus all other p-values should be taken with a pinch of salt.

Reading the article, there's several issues of reporting statistical analysis which are unclear, please address.

What covariates did you use in your models?

What method did you use to adjust for multiple comparisons?

lines 116-117 (Although there was no demonstrable between-group difference in response to apelin,

117 there appeared to be a trend towards a more prolonged response in those with CKD) are totally hearsay unless the actual numbers are presented. As a statistician I veto the use of the phrase "there is a trend towards" in all cases: it is poor interpretation of the statistical inference.

But I can't comment on what it is here since actual numbers aren't presented.

REVIEWER 1

I was pleased to be asked by the Editors of *Nature Communications* to review the paper 'Cardiovascular and renal effects of apelin in chronic kidney disease: A randomised, double-blind, placebo-controlled, crossover study'.

What are the noteworthy results?

Apelin exerts effects on blood pressure, systemic vascular resistance, increased cardiac index, increased renal blood flow and increased natriuresis and free water clearance; all of which has not been reported in such detail before.

Will the work be of significance to the field and related fields? How does it compare to the established literature? If the work is not original, please provide relevant references. This work is novel and no prior studies in the field have systematically examined effects of apelin on experimental endpoints as used by the authors.

Does the work support the conclusions and claims, or is additional evidence needed? The work presented herein supports the conclusions by the authors.

Are there any flaws in the data analysis, interpretation and conclusions? Do these prohibit publication or require revision? No.

Is the methodology sound? Does the work meet the expected standards in your field? Yes.

Is there enough detail provided in the methods for the work to be reproduced? Yes.

Thank you to the Reviewer for their encouraging comments and for their recognition of the novelty of our study.

1. The effect of apelin on systolic and diastolic blood pressure (Supplementary figure 4) seems to vanish at 20 minutes and there seems to be a compensatory increase thereafter. Do the authors have any explanations for this phenomenon?

As the Reviewer highlights, the effects of 30 nmol/min [Pyr¹]apelin-13 on blood pressure were short-lived and waned within 20 min. These data are consistent with previous studies from our group.¹ Here, a prolonged (6 hour) infusion of 30 nmol/min [Pyr¹]apelin-13 reduced mean arterial pressure in patients with heart failure by a maximum 4 mmHg (**Figure 1**), effects that were most apparent within the first 20 min of the apelin infusion. In this study, no significant change in mean arterial pressure was seen in healthy subjects. There was no evidence of receptor desensitisation, with persisting effects on other cardiovascular parameters (e.g., cardiac output and systemic vascular resistance).

[figure redacted]

Figure 1. Systemic infusion of 30 nmol/min [Pyr¹]apelin-13 (closed circles) and placebo (open circles) in patients with chronic heart failure.¹

In the current study, we found that apelin increased cardiac output and reduced systemic vascular resistance to a similar degree (by ~10% and ~13%, respectively). Blood pressure is a product of these indices and so it is perhaps unsurprising that the effect of apelin on blood pressure is modest. An acute reduction in blood pressure will trigger compensatory vasoconstrictor responses (e.g., activation of the renin-angiotensin and sympathetic nervous systems), which we hypothesise partly explain the observed short-lived effects.

The Reviewer also comments that there appears to be a compensatory increase in blood pressure following completion of the study infusion. In fact, we observed this in both healthy subjects and those with chronic kidney disease (CKD), in both placebo and apelin arms of the study. We have seen this in our previous studies with a similar design.^{2,3} These studies are long (~8 hours), with participants fasted and supine throughout. Often, they start to become restless towards the end. This may also reflect the waning of usual antihypertensive medication effects which all subjects take on the morning of the study. This is demonstrated in **Figure 2**, which shows mean arterial pressure of healthy subjects (blue) and patients with CKD (red) throughout the study course. The rise in blood pressure is most apparent at the end of the study day.

Figure 2. Mean arterial pressure throughout the course of the study day in healthy subjects (**A**, blue) and patients with CKD (**B**, red).

2. Data on heart rate would be interesting and authors might provide these in a supplemental file. How do the authors explain differences observed between healthy individuals and CKD patients? Does it correlate with arterial stiffness?

Thank you for highlighting the notable difference in heart rate response to apelin. Despite no differences in baseline heart rate (healthy subjects: 58 ± 2 bpm *versus* CKD 59 ± 2 bpm; $p=0.866$), apelin increased heart rate by 4 bpm in healthy subjects compared to baseline, whereas no response was seen in patients with CKD. Our results are in keeping with two of our own previous studies where heart rate increased in response to apelin in healthy subjects but not in patients with heart failure.^{1,4} We observed no relationship between heart rate and pulse wave velocity (our measure of arterial stiffness) in healthy subjects. However, the observed effect on heart rate was short-lived, returning to baseline within 20 min in contrast to the more prolonged fall in pulse wave velocity. These data are shown below (**Figure 3**) and have been added to the revised manuscript as Supplementary figure 6. We cannot fully explain the observed difference in response between groups. The minority of CKD patients (25%) were prescribed rate-limiting medications.

Figure 3. Heart rate increased in response to 30 nmol/min [Pyr¹]apelin-13 in healthy subjects (A) but not in patients with CKD (B). Analyses were by 2-way ANOVA or mixed effects model with Dunnett's test to adjust for multiple comparisons. *p<0.05 for within group comparisons to baseline.

3. Another interesting finding is that alterations to the blood flow seem to occur at a later stage in the CKD population. How do authors explain this? Does that eventually mean that full effects of apelin are somehow delayed in the CKD population?

As the Reviewer recognises, apelin increased renal blood flow but this effect appeared to be delayed in patients with CKD with the greatest effect occurring beyond the end of the infusion. This was similar for natriuresis and free water clearance but contrasted with the immediate systemic haemodynamic effects of apelin in this group. Notably, the raw data do suggest that there may be exaggerated and prolonged effects of apelin on cardiac output and systemic vascular resistance in patients with CKD compared to healthy subjects, but only cardiac output met statistical significance.

There are several possible explanations for the observed differences. First, it is likely that the apelin system is altered in CKD. There are no published human data, however animal models of CKD show downregulation of the system.⁵ CKD may also alter apelin receptor density within the glomerulus and between the afferent and efferent arterioles; these changes might influence the response to apelin. Human studies have demonstrated delayed renal responses to other vasoactive peptides (e.g., endothelin-1).⁶ Whilst some effects are an immediate consequence of receptor activation (such as nitric oxide generation and rapid vasodilatation), others rely on complex signalling cascades or alterations in other proteins (e.g., aquaporin 2 channels). Alterations in apelin receptor signalling in CKD may mean the effects of apelin take longer to switch 'on' or 'off'.

Second, the relationships between the apelin and renin-angiotensin systems are likely to influence the differential effects of apelin in health and CKD. It is recognised that there is crosstalk between these systems. In a model of heart failure, angiotensin II infusion downregulated apelin receptor expression; this was restored by an angiotensin receptor blocker.⁷ The renin-angiotensin system is upregulated in CKD and most patients in our study were prescribed either an ACE inhibitor or an angiotensin receptor blocker. We hypothesise that apelin signalling is altered by these medications. Additionally, inhibition of the renin-angiotensin system may limit the compensatory vasoconstrictor response to apelin in this group, potentially prolonging and enhancing its effect.

We would caution that we have examined the short-term effects of apelin in CKD. It is difficult to comment whether these would be maintained longer-term. However, our previous work using a similar approach in the endothelin field in CKD showed this to be the case^{2,3,8} and led to the recent approval of the combined angiotensin/endothelin blocker, sparsentan, for patients with IgA nephropathy.⁹ Importantly, the current data justify the need for further studies of apelin in CKD. We have added a paragraph to the revised Discussion covering these points.

4. It is of course speculative, but what kind of direct effects would the authors expect when discussing renal haemodynamics?

Thank you for this important question. Renal haemodynamics are tightly regulated and are influenced by systemic blood pressure, renal blood flow and changes within the renal cortex (glomerulus) and medulla. Within the glomerulus, this is primarily due to alterations in the diameter of the afferent/efferent glomerular arterioles as well as intraglomerular modifications that affect the filtration coefficient (such as mesangial cell contraction, basement membrane structure and charge). Within the medulla, changes in the vasa recta and tubular epithelial cells contribute. Clinical studies are only able to determine renal blood flow and renal clearance.

As discussed in our manuscript, based on the observed increase in renal blood flow, decreased glomerular filtration rate (GFR) and reduced filtration fraction, apelin appears to cause a relative efferent to afferent vasodilatory effect on the glomerular arterioles (lines 234-244). We hypothesise that apelin is also likely to have direct intraglomerular effects. Specifically, we think it may promote relaxation of mesangial cells (by opposing the actions of angiotensin II and possibly vasopressin). This would be expected to increase the filtration coefficient (which would increase GFR in the absence of changes in ultrafiltration pressure). However, this effect is likely to be modest and unlikely to counterbalance the effects of afferent and efferent vasodilatation.

Transforming growth factor- β 1 (TGF- β 1) promotes contraction of mesangial cells and production of mesangial matrix, ultimately leading to scarring and glomerulosclerosis. It also acts on tubular epithelial cells and is an important mediator of progressive injury and interstitial fibrosis. Preclinical data from models of kidney injury finds that apelin prevents injury-induced elevation of TGF- β 1 and can inhibit its effects.^{10,11} Inhibition of TGF- β 1 will not be of relevance in our acute studies, but whether apelin may prevent these damaging alterations in mesangial and tubular epithelial cells in the long term is of interest.

5. Not all patients were on RASi and as mentioned by the authors, only a minority on SGLT2i. Was there a difference in proteinuria reduction when no RASi was used or in those where SGLT2i were introduced?

Thank you for this question. Two patients commenced SGLT2i during the study period (for one patient, this occurred between study visits). They omitted these drugs for at least seven days prior to the study, therefore we cannot comment on whether they influenced proteinuria reduction in response to apelin. With respect to renin-angiotensin system inhibitors, 10 of 12 patients with CKD were prescribed either an ACE inhibitor or angiotensin receptor blocker (we apologise that there is an error in Table 1 which states 2 patients were prescribed an ARB – this should say three and has been corrected). The Reviewer will appreciate these numbers are too small to examine whether there was any association between prescription of a renin-angiotensin system inhibitor and proteinuria reduction, but we present the raw data below in **Figure 4**. This will be an important question for future clinical trials to address

Figure 4. Absolute reduction in protein excretion for each patient with CKD in response to 1 nmol/min and 30 nmol/min [Pyr¹]apelin-13. Patients who were not prescribed a renin-angiotensin system inhibitor are shown in red.

6. One typo: (p8, line 192) should be our, otherwise very well written.

Many thanks for pointing out this error which we have now corrected.

7. I slightly disagree when authors mention a homogeneous population and then mention vasculitis – overall, 3 patients had a vasculitis.

We thank the Reviewer for highlighting this and appreciate their opinion. In our study cohort, two patients had ANCA-associated vasculitis and 1 anti-glomerular basement membrane disease; the remainder had renal-specific disease. The three patients with multi-system disease were all in remission with no ongoing systemic inflammation and only two continued on immunosuppression. Our intention was to highlight that the included patients did not have active systemic inflammation, and we have now edited the text to reflect this.

8. Did you see a trend for more pronounced effects when apelin was used in more severe CKD stages, e.g. in those with a GFR of below 30?

Thank you for this interesting question. We examined the relationship between GFR and the systemic and renal effects of apelin but did not identify any associations. The Reviewer will appreciate that our study was not powered to identify significant differences in apelin effects across the stages of CKD. We have not commented on this in the text as we do not think this reflects the absence of a relationship but is more likely due to the modest sample size.

9. Methods: how was stable CKD defined?

We apologise that this important definition was not clear in the manuscript. Subjects with CKD were required to have at least two estimated GFR measures that were within 10% of each other over a 3-month period. If the subject had multi-system inflammatory disease this had to be in remission, defined by the treating clinician and confirmed by the investigators at the time of recruitment. Vasculitis was the only diagnosis in our CKD cohort that fell into this category. This has now been clarified in the Methods section of the revised manuscript.

10. Assays: the authors mention that apelin concentrations were measured by an ELISA, but such results are not given. The same is true for iohexol and PAH concentrations.

We apologise for this inconsistency. We did indeed measure plasma apelin concentration in our study. This confirmed that only the higher dose infusion (30 nmol/min) led to a significant increase in circulating apelin concentration (**Figure 5**). This was consistent with our hypothesis and was our intention. We have now added these new data as Supplementary figure 1 to the revised manuscript. Plasma iohexol and PAH concentrations were measured by high performance liquid chromatography with ultraviolet detection. The raw data are not presented but these were used to calculate glomerular filtration rate and renal blood flow, respectively. We have stated the relevant equations in the revised Methods section.

Figure 5. Circulating apelin concentration did not change with infusion of 1 nmol/min [Pyr¹]apelin-13 but rose significantly with infusion of 30 nmol/min [Pyr¹]apelin-13. Analysis by paired t-test. ** p<0.01 and **** p<0.001 for each group in comparison to their respective baselines.

11. Importantly, did any of the patients report any side effects when the low dose or the higher dose of apelin was administered?

Thank you for raising this important point and we apologise if this was not clear. No subject receiving low and high dose apelin experienced any side effects. We have now clarified this within the text.

REVIEWER 2

This manuscript presents a meticulously conducted clinical study focusing on a novel approach to treating CKD, showcasing exemplary methodological rigor. Despite its modest scale as a proof-of-concept investigation, it unveils compelling therapeutic prospects. The authors are recognized experts in the field with multiple publications on the subject.

We thank the Reviewer for their very positive comments. We are delighted that they consider our study to be '*meticulously conducted...showcasing exemplary methodological rigor*' and that they recognise the significance of our findings for future management of kidney disease.

1. In your previous work (Reference 34), you demonstrated higher apelin levels in women. Did you observe any sex differences in the effects of apelin within your study population?

Thank you for this very interesting question. In each study group (both healthy subjects and patients with CKD), 8 of 12 participants were male. **Figure 6** shows the effects of apelin in males and females for the primary endpoints of our study, As the Reviewer will appreciate our study was not powered to examine this therefore, we have not done any further analysis and do not intend to discuss this in our manuscript. It may be interesting to examine this in larger studies.

Figure 6. Differential effects of apelin in males and females for primary endpoints of systemic vascular resistance index (A – healthy subjects; B – CKD) and effective renal blood flow (C- healthy subjects; D- CKD).

2. In this same previous work (Reference 34), you demonstrated an inverse relationship between apelin levels and GFR. Looking at this finding, it seems reasonable to hypothesize a certain level of apelin resistance in patients with CKD. However, you show in this new study that certain effects are either more pronounced or prolonged in CKD, which is even more surprising knowing that apelin receptors are prone to desensitization and internalization. What are the authors thoughts on this matter? Could the pharmaceutical RAAS inhibition play a role here?

We thank the Reviewer for this excellent question. As recognised by the Reviewer, the apelin receptor undergoes internalisation and desensitisation. However, our current findings are in line with our earlier work where we infused 30 nmol/min [Pyr¹]apelin-13 for 6 hours to patients with heart failure.¹ In that study, apelin lowered mean arterial pressure and systemic vascular resistance index and increased cardiac index without evidence of receptor desensitization. We fully agree that concomitant renin-angiotensin system inhibition is likely to be implicated here. There is recognised crosstalk between the systems, and we hypothesise that apelin signaling is altered by angiotensin II inhibition. We have added the following text to the revised Discussion focusing on the different responses observed in health and CKD:

The effects of apelin on renal parameters appeared delayed in patients with CKD in comparison to healthy subjects, and there was a suggestion of prolonged and perhaps increased systemic hemodynamic effects in this group. There are several potential explanations for these differences. The apelin system is expressed throughout the human kidney¹² and whilst there are no published human data regarding the effects of CKD on the system, it is downregulated in animal models.⁵ We expect the system will be altered in humans with CKD. The relationship between the apelin system and renin-angiotensin system is also implicated. There is cross-talk between the systems: for example, in a model of heart failure angiotensin II-induced downregulation of the apelin receptor is restored by angiotensin receptor blockade.⁷ In our study, most patients with CKD were established on renin-angiotensin system blockers and these drugs may alter apelin signaling. Inhibition of angiotensin II may also affect the speed at which compensatory mechanisms are able to respond to acute changes in systemic hemodynamics.

3. In this same previous work (Reference 34), you demonstrated a colocalization of apelin, apelin receptor and SGLT2 in the kidneys, and you now show that apelin and SGLT2 inhibitors share certain effects, e.g., on proteinuria.

a. Do you think that there might be a crosstalk between both pathways?

Thank you for this insightful question. As recognised by the Reviewer, the effects of apelin and sodium-glucose cotransporter 2 (SGLT2) inhibitors on cardiovascular and renal endpoints are similar. This is interesting given the colocalization of the apelin receptors and SGLT2 within the proximal convoluted tubule (see **Figure 7** below).¹² We agree that there may be a functional interaction between the two. This is supported by *in vitro* studies of diabetic human coronary artery smooth muscle cells and endothelial cells, where apelin expression was increased in response to empagliflozin.¹³ It is our intention to examine this further in future clinical studies.

[figure redacted]

Figure 7. Localization of apelin (**A-D**), ELA (**E-H**) and apelin receptor (**I-L**) protein in the proximal convoluted tubule (PCT) within the cortex. Aquaporin 1 (AQP1; **C, G, K**) is expressed by proximal tubular epithelial cells and acts as a marker of the PCT. Apelin receptor (**M-P**) protein also colocalizes with the sodium-glucose cotransporter 2 (SGLT2; **O**) protein within the PCT. Scale bar = 50µm. Reproduced from Nyimanu & Chapman, *Brit J Clin Pharm* 2022

b. The natriuresis induced by SGLT2 inhibitors has been shown to be transient (Rao et al., JASN, 2023). Do you think that the natriuresis induced by apelin might also be transient?

An interesting question. The natriuretic effect of apelin was significant and suggests a direct tubular effect as one would expect the kidney to retain salt and water in response to the apelin-induced fall in blood pressure and glomerular filtration rate. Our data do not allow us to localise this response to a particular nephron segment/salt transporter. However, we agree with the Reviewer's comment that this may well be a transient effect due to compensatory changes in salt transport within the nephron. As discussed above, we also hypothesise that there may be functional interaction with the SGLT2. Future studies using long-acting apelin receptor analogues will allow us to examine this in more detail.

4. Concerning the two different doses of apelin, you refer to two publications with dose-response experiments. Some more information in the manuscript would be helpful for a better understanding of the choice of these two different doses. Also, some comment would be useful about what dose would be most suitable in the opinion of the authors for further evaluation in CKD?

Thank you. We have now added more details to the revised manuscript regarding the apelin doses used. In addition to our previous published studies in healthy subjects and patients with heart failure,^{1,4,14} our local vascular studies in patients with CKD found a dose of 1 nmol/min [Pyr¹]apelin-13 caused vasodilatation without discernible systemic effects whereas 30 nmol/min had both local and systemic effects on vascular tone (**Figure 8**). In the current study, we wanted to use a sub-systemic dose to determine if apelin might have direct renal effects without confounding by systemic vascular effects (e.g., a fall in blood pressure which would trigger renal conservation of salt and water). A systemic dose would be most appropriate for future clinical studies in CKD as the higher dose used here had both cardioprotective (improved systemic vascular resistance and cardiac output) and renoprotective (reduced filtration fraction and natriuresis / free water clearance) effects. Ideally, future studies should be performed with a long-acting oral apelin analogue, and the current data will help in their design in terms of dose.

A. Healthy volunteers

B. Chronic kidney disease

C.

Figure 8. Forearm blood flow studies in healthy subjects (**A**; blue) and patients with CKD (**B**; red). Incremental doses of [Pyr¹]apelin-13 were infused *via* the left brachial artery (closed circles; infused arm); the right arm acted as a contemporaneous control (open circles; non-infused arm). Apelin led to a dose-dependent increase in forearm blood flow in both groups. There was no difference in response between groups when comparing the ratio of blood flow between arms (**C**). Spillover into the systemic circulation was seen with doses >3 nmol/min. Data shown are mean±SEM. Analysis by 2-way ANOVA (health) and mixed-effects model (CKD and ratio comparisons) with Šidák's multiple comparison test. Between arm comparison * p<0.05; ** p<0.01; *** p<0.001; **** p<0.0001.

5. Can you hypothesize why an increase in heart rate is observed in healthy subjects but not in CKD patients?

Thank you for highlighting the notable difference in heart rate response to apelin which was also noted by Reviewer 1 (comment 2). Despite no differences in baseline heart rate (healthy subjects: 58 ± 2 bpm versus CKD 59 ± 2 bpm; $p=0.866$), apelin increased heart rate by 4 bpm in healthy subjects compared to baseline, whereas no response was seen in patients with CKD. Our results are in keeping with two of our own previous studies where heart rate increased in response to apelin in healthy subjects but not in patients with heart failure.^{1,4} However, the observed effect on heart rate was short-lived, returning to baseline within 20 min. These data are shown below (**Figure 3, see p3, Response to Reviewer 1**) and have now been added to the revised manuscript as Supplementary figure 5. We cannot fully explain the observed difference in response between groups. The minority of CKD patients (25%) were prescribed rate-limiting medications.

Figure 3. Heart rate increased in response to 30 nmol/min [Pyr¹]apelin-13 in healthy subjects (A) but not in patients with CKD (B). Analyses were by 2-way ANOVA or mixed effects model with Dunnett's test to adjust for multiple comparisons. * $p < 0.05$ for within group comparisons to baseline.

6. You discussed the interplay between apelin and angiotensin II and arginine vasopressin (AVP). You excluded patients with ADPKD treated with tolvaptan to avoid interference with AVP activity. However, most of your CKD patients were treated with either an angiotensin-converting enzyme inhibitor or an angiotensin receptor blocker. Do you believe this could influence the impact of apelin on your study outcomes?

Thank you for this important question. As discussed in response to comment 2, we agree that renin-angiotensin system blockers are likely to influence the response to apelin. Our study aimed to examine the potential cardioprotective and renoprotective effects of apelin in patients with CKD. Apelin as a potential treatment for CKD is almost always going to be as add-on to standard-of-care. Renin-angiotensin system inhibitors are the cornerstone of CKD management therefore any study needs to demonstrate beneficial effects in addition to these.

7. You may want to consider discussing certain limitations in implementing apelin analogues for treating CKD.

a. The decrease in blood pressure appears to be primarily driven by the reduction in diastolic blood pressure. Why do you think this is the case, and do you believe it could limit its use in patients with coronary heart disease, considering that coronary blood flow greatly depends on diastolic pressure?

We appreciate the Reviewer raising this important point. With respect to the relative reduction in diastolic blood pressure compared to systolic blood pressure, it is likely that this represents the fall in systemic vascular resistance. Overall, the reduction in diastolic blood pressure remains modest and, in CKD, apelin reduced diastolic blood pressure similarly to that seen in healthy subjects at baseline, so we do not believe this will compromise coronary perfusion.

b. As you mentioned, apelin is rather thought to act in a paracrine or autocrine manner. Systemic administration of long acting apelin analogues inevitably raises concerns about selectivity. Of particular concern is the angiogenic effect of apelin, especially in oncology patients and those with diabetes mellitus and diabetic retinopathy.

Thank you for highlighting this. Concerns regarding the potential angiogenic actions of apelin agonists and how these may be mitigated has been considered in a recent review.¹⁵ Here, the authors suggest that for peptide agonists targeting the apelin receptor in cardiovascular disease, selectivity could be achieved using engineered nanotechnologies as drug delivery systems. Such a development has been reported for [Pyr¹]apelin-13.¹⁶ The authors also highlight the rarity of cardiac tumours and thus the strategy of using targeted in situ delivery may be explored to allow for safer administration of apelin agonists drugs. Alternatively, local administration using biomaterials such as hydrogel might limit off target systemic actions of apelin agonists drugs.¹⁷⁻¹⁹

8. In your introduction, you mentioned another apelin receptor ligand 'elabela'. Would you anticipate that elabela might have a different effect profile compared to apelin?

We appreciate the Reviewer raising this. ELA is the critical mediator in development, with apelin being detectable at later stages. In our previous work, we have shown that both peptides are present in the adult human cardiovascular system, mediate comparable effects *in vivo* and can both be blocked by apelin receptor antagonists.²⁰ Predicted ELA isoforms and apelin activate downstream signalling pathways with cardiovascular relevance, specifically inhibition of cAMP formation, phosphorylation of ERK1/2, phosphorylation of nitric oxide synthase, β -arrestin recruitment and receptor internalisation. In all these assays, the effects of apelin and ELA peptides are comparable with no difference in downstream signalling. We also reported cardiovascular metrics such as blood pressure, cardiac hypertrophy, heart contractility in animals, and actions *in vitro* cultured vascular endothelial cells.²⁰ Again, in all these studies, the actions of apelin and ELA were comparable and we would anticipate similar results in clinical studies.

REVIEWER 4

1. Sound research and very clear write up. Promising results which I hope will be confirmed in future phase II/III studies. I comment as a statistician with no expertise on the medical background/discussion.

We are grateful to the statistical Reviewer for their positive comments, and we are delighted that they found our manuscript to be clear.

2. The study appears to be well-powered. But why was a power of 85% chosen? Given the relatively small sample size you could have aimed for 90% power. What was your rationale? Please explain.

Thank you for raising this point. Our study was originally designed in 2018-2019. However, the protocol was not finalised until the early months of the COVID-19 pandemic. Whilst we originally intended to aim for 90% power, we were conscious that the pandemic would be an ongoing concern for the duration of the study period. Patients with CKD remain at significantly increased risk from COVID-19, and this was particularly relevant in the pre-vaccine era when they were shielding. We appreciated that we might face significant challenges with recruitment due to understandable reluctance to attend hospital for non-essential visits. We therefore made the decision to aim for 85% power to ensure we were able to meet our recruitment target rather than risk being under-powered for our results. Whilst we acknowledge the sample size appears small, these are long and difficult study visits for participants and powering at 90% would have required us to recruit an additional three subjects per group, with a minimum of two visits each.

3. I like the figures/visualisations, however, as a statistician I can't recommend publication without a more detailed reporting of the outcomes of the ANOVA and mixed models. I want to see 95% confidence intervals and SE.

We are pleased that the Reviewer liked our figures. We apologise for not providing sufficient detail in our statistical reporting. There are several reasons why we did not include 95% confidence intervals and standard errors for all ANOVA/mixed model analyses in the manuscript. We appreciate that we are presenting a large amount of data and we have tried to ensure these are as clear as possible for the broad readership of the Journal. We are keen to avoid making the text too 'number heavy' and believe it is important to present results in context for clarity, accepting that some readers might be unfamiliar with the specific endpoints examined. We have provided the relevant absolute numerical change (with baseline data presented in Table 2). For example, '*In healthy subjects, pulse wave velocity fell by ~5% in response to [Pyr¹]apelin-13 30 nmol/min, (6.3±0.2 to 5.9±0.2 m/s, p<0.01 compared to placebo)*'. For the same reason, our figures do not show the raw data but instead the placebo-corrected percentage change from baseline, as we felt this was more understandable. The source data will be provided in an accompanying Excel document to the manuscript. However, we have provided model outputs for the results presented in the figures in **Appendix 1** for the Reviewer. We would prefer to leave the figures as presented but will be guided by the Editor.

4. You are only powered for null-hypothesis testing for the primary endpoint. Thus, all other p-values should be taken with a pinch of salt.

We agree with the Reviewer and have now commented on this specifically within the limitations section of the revised manuscript.

5. Reading the article, there are several issues of reporting statistical analysis which are unclear, please address.

a. What covariates did you use in your models?

We apologise for any unclear statistical reporting in our manuscript. For each ANOVA or mixed effects model, we examined changes in the variable of interest using the covariates of time (prespecified timepoints), participant group (healthy subjects or patients with chronic kidney disease), and treatment (apelin or placebo). We have now clarified this in the revised Methods section of the manuscript.

b. What method did you use to adjust for multiple comparisons?

Where a statistical model identified a significant result, we proceeded to multiple comparisons within each group – healthy subjects and patients with CKD – correcting for multiple comparisons to baseline using Dunnett's test, and between groups using Šidák's test. We have now stated this more clearly in the revised manuscript.

c. Lines 116-117 (Although there was no demonstrable between-group difference in response to apelin, there appeared to be a trend towards a more prolonged response in those with CKD) are totally hearsay unless the actual numbers are presented. As a statistician I veto the use of the phrase 'there is a trend towards' in all cases: it is poor interpretation of the statistical inference. But I can't comment on what it is here since actual numbers aren't presented.

We agree with the Reviewer's comments and apologise for the clumsy language. We have removed the phrase 'there is a trend towards'. As discussed in our response to comment 3, we appreciate the complexity of our studies and that we present a lot of data. Our goal is to ensure that our manuscript is clear, interesting and as accessible as possible to the reader. This is our rationale for highlighting key results within the text and choosing the best representations of the data for the figures, rather than presenting all the raw data. Hopefully, we have now improved our statistical communication in the revised manuscript.

References

1. Barnes GD, Alam S, Carter G, Pedersen CM, Lee KM, Hubbard TJ, Veitch S, Jeong H, White A, Cruden NL, et al. Sustained cardiovascular actions of APJ agonism during renin-angiotensin system activation and in patients with heart failure. *Circ Heart Fail*. 2013;6:482-491. doi: 10.1161/CIRCHEARTFAILURE.111.000077
2. Dhaun N, Macintyre IM, Melville V, Lilitkarntakul P, Johnston NR, Goddard J, Webb DJ. Blood pressure-independent reduction in proteinuria and arterial stiffness after acute endothelin-a receptor antagonism in chronic kidney disease. *Hypertension*. 2009;54:113-119. doi: 10.1161/HYPERTENSIONAHA.109.132670
3. Dhaun N, MacIntyre IM, Kerr D, Melville V, Johnston NR, Haughie S, Goddard J, Webb DJ. Selective endothelin-A receptor antagonism reduces proteinuria, blood pressure, and arterial stiffness in chronic proteinuric kidney disease. *Hypertension*. 2011;57:772-779. doi: 10.1161/HYPERTENSIONAHA.110.167486
4. Japp AG, Cruden NL, Barnes G, van Gemeren N, Mathews J, Adamson J, Johnston NR, Denvir MA, Megson IL, Flapan AD, et al. Acute cardiovascular effects of apelin in humans: potential role in patients with chronic heart failure. *Circulation*. 2010;121:1818-1827. doi: 10.1161/CIRCULATIONAHA.109.911339
5. Najafipour H, Soltani Hekmat A, Nekooian AA, Esmaeili-Mahani S. Apelin receptor expression in ischemic and non- ischemic kidneys and cardiovascular responses to apelin in chronic two-kidney-one-clip hypertension in rats. *Regul Pept*. 2012;178:43-50. doi: 10.1016/j.regpep.2012.06.006
6. Hunter RW, Moorhouse R, Farrah TE, MacIntyre IM, Asai T, Gallacher PJ, Kerr D, Melville V, Czopek A, Morrison EE, et al. First-in-Man Demonstration of Direct Endothelin-Mediated Natriuresis and Diuresis. *Hypertension*. 2017;70:192-200.
7. Iwanaga Y, Kihara Y, Takenaka H, Kita T. Down-regulation of cardiac apelin system in hypertrophied and failing hearts: Possible role of angiotensin II-angiotensin type 1 receptor system. *J Mol Cell Cardiol*. 2006;41:798-806. doi: 10.1016/j.yjmcc.2006.07.004
8. Goddard J, Johnston NR, Hand MF, Cumming AD, Rabelink TJ, Rankin AJ, Webb DJ. Endothelin-A receptor antagonism reduces blood pressure and increases renal blood flow in hypertensive patients with chronic renal failure: a comparison of selective and combined endothelin receptor blockade. *Circulation*. 2004;109:1186-1193.
9. Rovin BH, Barratt J, Heerspink HJL, Alpers CE, Bieler S, Chae DW, Diva UA, Floege J, Gesualdo L, Inrig JK, et al. Efficacy and safety of sparsentan versus irbesartan in patients with IgA nephropathy (PROTECT): 2-year results from a randomised, active-controlled, phase 3 trial. *Lancet*. 2023;402:2077-2090. doi: 10.1016/s0140-6736(23)02302-4
10. Chen H, Wan D, Wang L, Peng A, Xiao H, Petersen RB, Liu C, Zheng L, Huang K. Apelin protects against acute renal injury by inhibiting TGF-beta1. *Biochim Biophys Acta*. 2015;1852:1278-1287. doi: 10.1016/j.bbadis.2015.02.013

11. Wang LY, Diao ZL, Zheng JF, Wu YR, Zhang QD, Liu WH. Apelin attenuates TGF-beta1-induced epithelial to mesenchymal transition via activation of PKC-epsilon in human renal tubular epithelial cells. *Peptides*. 2017;96:44-52. doi: 10.1016/j.peptides.2017.08.006
12. Nyimanu D, Chapman FA, Gallacher PJ, Kuc RE, Williams TL, Newby DE, Maguire JJ, Davenport AP, Dhaun N. Apelin is expressed throughout the human kidney, is elevated in chronic kidney disease & associates independently with decline in kidney function. *Br J Clin Pharmacol*. 2022. doi: 10.1111/bcp.15446
13. Dutzmann J, Bode LM, Kalies K, Korte L, Knöpp K, Kloss FJ, Sirisko M, Pilowski C, Koch S, Schenk H, et al. Empagliflozin prevents neointima formation by impairing smooth muscle cell proliferation and accelerating endothelial regeneration. *Front Cardiovasc Med*. 2022;9:956041. doi: 10.3389/fcvm.2022.956041
14. Japp AG, Cruden NL, Amer DA, Li VK, Goudie EB, Johnston NR, Sharma S, Neilson I, Webb DJ, Megson IL, et al. Vascular effects of apelin in vivo in man. *J Am Coll Cardiol*. 2008;52:908-913. doi: 10.1016/j.jacc.2008.06.013
15. Rossin D, Vanni R, Lo Iacono M, Cristallini C, Giachino C, Rastaldo R. APJ as Promising Therapeutic Target of Peptide Analogues in Myocardial Infarction- and Hypertension-Induced Heart Failure. *Pharmaceutics*. 2023;15. doi: 10.3390/pharmaceutics15051408
16. Serpooshan V, Sivanesan S, Huang X, Mahmoudi M, Malkovskiy AV, Zhao M, Inayathullah M, Wagh D, Zhang XJ, Metzler S, et al. [Pyr1]-Apelin-13 delivery via nano-liposomal encapsulation attenuates pressure overload-induced cardiac dysfunction. *Biomaterials*. 2015;37:289-298. doi: 10.1016/j.biomaterials.2014.08.045
17. Fang J, Koh J, Fang Q, Qiu H, Archang MM, Hasani-Sadrabadi MM, Miwa H, Zhong X, Sievers R, Gao DW, et al. Injectable Drug-Releasing Microporous Annealed Particle Scaffolds for Treating Myocardial Infarction. *Adv Funct Mater*. 2020;30. doi: 10.1002/adfm.202004307
18. Mitchell MJ, Billingsley MM, Haley RM, Wechsler ME, Peppas NA, Langer R. Engineering precision nanoparticles for drug delivery. *Nat Rev Drug Discov*. 2021;20:101-124. doi: 10.1038/s41573-020-0090-8
19. Almas T, Haider R, Malik J, Mehmood A, Alvi A, Naz H, Satti DI, Zaidi SMJ, AlSubai AK, AlNajdi S, et al. Nanotechnology in interventional cardiology: A state-of-the-art review. *Int J Cardiol Heart Vasc*. 2022;43:101149. doi: 10.1016/j.ijcha.2022.101149
20. Yang P, Read C, Kuc RE, Buonincontri G, Southwood M, Torella R, Upton PD, Crosby A, Sawiak SJ, Carpenter TA, et al. Elabela/Toddler Is an Endogenous Agonist of the Apelin APJ Receptor in the Adult Cardiovascular System, and Exogenous Administration of the Peptide Compensates for the Downregulation of Its Expression in Pulmonary Arterial Hypertension. *Circulation*. 2017;135:1160-1173. doi: 10.1161/CIRCULATIONAHA.116.023218

APPENDIX 1

i. Placebo-corrected percentage change in mean arterial pressure from baseline (Fig 1A)

[Pyr¹]apelin-13 30 nmol/min

Mixed effects model with Dunnett's multiple comparisons test

Comparison	Predicted (LS) mean difference	95% CI	SE of diff.	P value
Healthy subjects				
0 vs 5 mins	-4.0	-7.8 to -0.1	1.5	0.045
0 vs 10 mins	-2.8	-6.4 to 0.9	1.4	0.211
0 vs 20 mins	2.3	-1.6 to 6.2	1.5	0.449
0 vs 30 mins	0.7	-2.9 to 4.2	1.3	0.994
0 vs 45 mins	1.5	-2.0 to 5.1	1.3	0.746
0 vs 60 mins	1.6	-1.9 to 5.1	1.3	0.717
CKD				
0 vs 5 mins	-4.2	-8.0 to -0.3	1.5	0.027
0 vs 10 mins	-1.7	-5.2 to 1.6	1.3	0.567
0 vs 20 mins	0.6	-3.2 to 4.4	1.5	0.997
0 vs 30 mins	-0.3	-3.7 to 3.0	1.3	>0.999
0 vs 45 mins	1.2	-2.2 to 4.6	1.3	0.861
0 vs 60 mins	-0.2	-3.6 to 3.2	1.3	>0.999

ii. Placebo-corrected percentage change in systemic vascular resistance index (Fig 1B)

[Pyr¹]apelin-13 30 nmol/min

Mixed effects model with Dunnett's multiple comparisons test

Comparison	Predicted (LS) mean difference	95% CI	SE of diff.	P value
Healthy subjects				
0 vs 5 mins	-12.0	-19.2 to -4.7	2.2	0.004
0 vs 10 mins	-8.9	-16.7 to -1.1	2.5	0.025
0 vs 20 mins	0.8	-11.6 to 13.3	3.7	>0.999
0 vs 30 mins	-2.1	-12.2 to 8.0	3.3	0.968
0 vs 45 mins	3.2	-7.2 to 13.8	3.4	0.849
0 vs 60 mins	4.0	-5.7 to 13.6	3.1	0.658
CKD				
0 vs 5 mins	-13.9	-30.7 to 2.9	4.5	0.098
0 vs 10 mins	-13.0	-23.8 to -2.2	3.5	0.018
0 vs 20 mins	-4.8	-17.9 to 8.5	3.5	0.633
0 vs 30 mins	-9.7	-20.9 to 1.6	3.6	0.100
0 vs 45 mins	-1.4	-16.2 to 13.4	4.8	>0.999
0 vs 60 mins	-1.8	-13.9 to 10.2	3.9	0.993

iii. Placebo-corrected percentage change in cardiac index (Fig 1C)

[Pyr¹]apelin-13 30 nmol/min

Mixed effects model with Dunnett's multiple comparisons test

Comparison	Predicted (LS) mean difference	95% CI	SE of diff.	P value
Healthy subjects				
0 vs 5 mins	10.7	0.3 to 21.1	4.0	0.043
0 vs 10 mins	7.7	-1.7 to 17.2	3.6	0.154
0 vs 20 mins	3.2	-7.3 to 13.6	4.0	0.929
0 vs 30 mins	3.5	-6.0 to 12.9	3.6	0.854
0 vs 45 mins	-0.9	-10.3 to 8.6	3.6	>0.999
0 vs 60 mins	-2.0	-11.5 to 7.4	3.6	0.986
CKD				
0 vs 5 mins	9.9	-1.6 to 21.3	4.4	0.122
0 vs 10 mins	14.2	4.5 to 23.9	3.7	0.001
0 vs 20 mins	6.9	-4.6 to 18.4	4.4	0.447
0 vs 30 mins	11.5	2.1 to 20.9	3.6	0.009
0 vs 45 mins	5.5	-3.9 to 14.9	3.6	0.464
0 vs 60 mins	2.4	-7.0 to 11.8	3.6	0.971

iv. Placebo-corrected percentage change in pulse wave velocity (Fig 1D)

[Pyr¹]apelin-13 30 nmol/min

2-way ANOVA with Dunnett's multiple comparisons test

Comparison	Mean difference	95% CI	SE of diff.	P value
Healthy subjects				
0 vs 15 mins	-6.9	-14.3 to 0.6	2.7	0.073
0 vs 45 mins	-8.4	-15.7 to -2.0	2.4	0.011
0 vs 75 mins	0.4	-7.5 to 8.4	2.9	0.998
CKD				
0 vs 15 mins	-0.2	-9.1 to 8.8	3.3	>0.999
0 vs 45 mins	-0.3	-8.1 to 7.5	2.9	0.999
0 vs 75 mins	1.9	-7.4 to 11.4	3.5	0.898

v. Placebo-corrected percentage change in effective renal blood flow from baseline (Fig 2A)

[Pyr¹]apelin-13 1 nmol/min

Mixed effects model with Dunnett's multiple comparison test

Comparison	Predicted (LS) mean difference	95% CI	SE of diff.	P value
Healthy subjects				
0 vs 30 mins	16.3	6.0 to 26.7	4.5	0.002
0 vs 60 mins	6.2	-4.2 to 16.5	4.5	0.298
CKD				
0 vs 30 mins	9.9	0.2 to 19.6	4.2	0.045
0 vs 60 mins	10.2	1.0 to 19.6	4.1	0.034

[Pyr¹]apelin-13 30 nmol/min

Mixed effects model with Dunnett's multiple comparison test

Comparison	Predicted (LS) mean difference	95% CI	SE of diff.	P value
Healthy subjects				
0 vs 30 mins	16.8	0.4 to 33.2	7.1	0.044
0 vs 60 mins	1.6	-14.8 to 17.9	7.1	0.964
CKD				
0 vs 30 mins	8.3	-6.7 to 23.3	6.5	0.349
0 vs 60 mins	21.2	5.8 to 36.5	6.7	0.006

vi. Placebo-corrected percentage change in glomerular filtration rate from baseline (Fig 2B)

[Pyr¹]apelin-13 1 nmol/min

Mixed effects model with Dunnett's multiple comparison test

Comparison	Predicted (LS) mean difference	95% CI	SE of diff.	P value
Healthy subjects				
0 vs 30 mins	7.0	-1.9 to 16.0	3.9	0.141
0 vs 60 mins	4.1	-5.2 to 13.3	4.0	0.505
CKD				
0 vs 30 mins	-9.3	-17.6 to -0.9	3.6	0.028
0 vs 60 mins	-6.4	-14.6 to 1.8	3.6	0.143

[Pyr¹]apelin-13 30 nmol/min

2-way ANOVA with Dunnett's multiple comparison test

Comparison	Mean difference	95% CI	SE of diff.	P value
Healthy subjects				
0 vs 30 mins	1.5	-16.0 to 19.1	6.7	0.964
0 vs 60 mins	-7.5	-23.9 to 8.1	6.0	0.389
CKD				
0 vs 30 mins	-14.4	-22.0 to -6.7	3.0	0.001
0 vs 60 mins	-5.1	-13.0 to 2.9	3.1	0.224

vii. Placebo-corrected absolute change in effective filtration fraction from baseline (Fig 2C)

[Pyr¹]apelin-13 1 nmol/min

Mixed effects model with Dunnett's multiple comparison test

Comparison	Predicted (LS) mean difference	95% CI	SE of diff.	P value
Healthy subjects				
0 vs 30 mins	-1.5	-3.2 to 0.2	0.7	0.097
0 vs 60 mins	-0.1	-1.8 to 1.7	0.8	0.992
CKD				
0 vs 30 mins	-3.0	-4.6 to -1.4	0.7	0.0002
0 vs 60 mins	-3.3	-4.8 to -1.7	0.7	<0.0001

[Pyr¹]apelin-13 30 nmol/min

Mixed effects model with Dunnett's multiple comparison test

Comparison	Predicted (LS) mean difference	95% CI	SE of diff.	P value
Healthy subjects				
0 vs 30 mins	-2.4	-4.6 to -0.1	2.3	0.045
0 vs 60 mins	-1.5	-3.8 to 0.9	1.5	0.264
CKD				
0 vs 30 mins	-4.2	-6.3 to -2.1	4.6	<0.0001
0 vs 60 mins	-5.2	-7.4 to -3.0	5.5	<0.0001

viii. Absolute change in protein excretion from baseline in CKD (Fig 2D)

[Pyr¹]apelin-13 1 nmol/min

Mixed effects model with Šidák's multiple comparison test

Comparison	Predicted (LS) mean difference	95% CI	SE of diff.	P value
Placebo				
0 vs 30 mins	18.6	-49.9 to 87.0	27.0	0.723
0 vs 60 mins	-17.8	-39.5 to 3.8	8.5	0.107
Apelin				
0 vs 30 mins	-57.3	-98.4 to -16.2	16.0	0.009
0 vs 60 mins	-47.4	-77.4 to -17.4	11.7	0.004

[Pyr¹]apelin-13 30 nmol/min

Mixed effects model with Šidák's multiple comparison test

Comparison	Predicted (LS) mean difference	95% CI	SE of diff.	P value
Placebo				
0 vs 30 mins	6.3	-29.9 to 42.3	15.6	0.903
0 vs 60 mins	-6.7	-42.9 to 29.5	15.6	0.891
Apelin				
0 vs 30 mins	-51.2	-87.4 to -15.1	15.6	0.004
0 vs 60 mins	-34.0	-71.6 to 3.12	16.0	0.078

ix. Placebo-corrected percentage change in urine sodium excretion (Fig 3A)

[Pyr¹]apelin-13 1 nmol/min

Mixed effects model with Dunnett's multiple comparison test

Comparison	Predicted (LS) mean difference	95% CI	SE of diff.	P value
Healthy subjects				
0 vs 30 mins	32.2	13.9 to 50.1	8.0	0.001
0 vs 60 mins	7.6	-12.0 to 27.2	8.5	0.583
CKD				
0 vs 30 mins	18.5	1.4 to 35.7	7.5	0.033
0 vs 60 mins	26.9	10.2 to 43.6	7.3	0.001

[Pyr¹]apelin-13 30 nmol/min

Mixed effects model with Dunnett's multiple comparison test

Comparison	Predicted (LS) mean difference	95% CI	SE of diff.	P value
Healthy subjects				
0 vs 30 mins	14.4	-4.2 to 33.0	7.1	0.127
0 vs 60 mins	11.9	-7.1 to 30.9	7.3	0.230
CKD				
0 vs 30 mins	10.9	6.9 to 28.6	7.0	0.249
0 vs 60 mins	28.6	9.5 to 47.6	7.4	0.006

x. Placebo-corrected percentage change in free water clearance (Fig 3B)

[Pyr¹]apelin-13 1 nmol/min

Mixed effects model with Dunnett's multiple comparison test

Comparison	Mean difference	95% CI	SE of diff.	P value
Healthy subjects				
0 vs 30 mins	31.4	2.2 to 60.6	10.9	0.037
0 vs 60 mins	12.4	-12.4 to 37.2	9.0	0.344
CKD				
0 vs 30 mins	9.9	1.1 to 18.6	3.4	0.029
0 vs 60 mins	14.2	2.5 to 25.8	4.6	0.019

[Pyr¹]apelin-13 30 nmol/min

Mixed effects model with Dunnett's multiple comparison test

Comparison	Mean difference	95% CI	SE of diff.	P value
Healthy subjects				
0 vs 30 mins	10.5	-16.6 to 37.5	10.1	0.547
0 vs 60 mins	12.4	-17.6 to 42.5	10.9	0.493
CKD				
0 vs 30 mins	4.7	-13.1 to 22.6	6.8	0.753
0 vs 60 mins	19.9	2.5 to 37.4	6.4	0.028

REVIEWERS' COMMENTS

Reviewer #1 (Remarks to the Author):

The authors have answered all my questions raised during the initial review satisfactory. There are some limitations which we acknowledge, but the research is very sound and the presented results compelling.

Reviewer #2 (Remarks to the Author):

The authors have answered all our questions, nothing more to add, this paper can now be accepted for publication.

Reviewer #3 (Remarks to the Author):

Reviewer #4 (Remarks to the Author):

Thank you for addressing all my questions.

Regarding the power of the study, I completely understand your rationale and given the circumstances (COVID) it was wise to prioritise completing recruitment at the expense of a 5% loss of power. In the interest of historical context, I would add a one line to the manuscript, perhaps under limitations, saying that a higher power 90% was feasible and more desirable but COVID restrictions made it more prudent to focus on completing recruitment in a COVID-vulnerable population. The point is not a “true” limitation, merely an acknowledgement of the difficult clinical circumstances under which the team conducted the research. It gives context to your choice of power, and demonstrates pragmatism and prudence.

Thank you for providing an Appendix with the outputs of the ANOVA and mixed models. This is very helpful to contextualise the data you present. I strongly advise, in fact I think it should be mandatory, that it is included in the final manuscript or added as supplemental material.

I appreciate your choice of presenting placebo-corrected percentage change from baseline in your figures for readability and ease of communication to a clinical audience. However, I disagree with you with regards to omitting the full details of the statistical modelling. It is one of the CONSORT guidelines: “For each primary and secondary outcome, results for each group, and the estimated effect size and its precision (such as 95% confidence interval)”. This should really be minimum standard requirement in the reporting of RCT trials.

Please see: <https://www.goodreports.org/reporting-checklists/consort/info/#17a> for further detail.

You say: “We are keen to avoid making the text too ‘number heavy’ and believe it is important to present results in context for clarity, accepting that some readers might be unfamiliar with the specific endpoints examined”, but from my prospective ‘number heavy’ is including too many p-values of secondary/exploratory endpoints while omitting crucial outputs of your primary endpoint’s statistical modelling!

I can do with fewer p-values and a greater focus on the primary endpoint. Hence my suggestion of taking p-values off the figures. Figures speak for themselves. Happy for the editorial guidelines to have final say.

Thank you for adding details of covariates used in the methods section (again this is standard practice reporting an RCT) and of corrections for multiple comparisons (ditto). This will add scientific rigour to your paper.

RESPONSE TO REVIEWERS

Reviewer 1

The authors have answered all my questions raised during the initial review satisfactory. There are some limitations which we acknowledge, but the research is very sound and the presented results compelling.

We are once again grateful to the Reviewer for their comments on our manuscript and are delighted we have been able to answer their questions. We appreciate that they feel our research is “very sound” with compelling results.

Reviewer 2

The authors have answered all our questions, nothing more to add, this paper can now be accepted for publication.

Thank you – we are delighted the Reviewer has recommended our manuscript is published!

Reviewer 3

We appreciate the Reviewer taking the time to co-review our manuscript and are grateful for their feedback.

Reviewer 4

Thank you for addressing all my questions.

Regarding the power of the study, I completely understand your rationale and given the circumstances (COVID) it was wise to prioritise completing recruitment at the expense of a 5% loss of power. In the interest of historical context, I would add a one line to the manuscript, perhaps under limitations, saying that a higher power 90% was feasible and more desirable but COVID restrictions made it more prudent to focus on completing recruitment in a COVID-vulnerable population. The point is not a “true” limitation, merely an acknowledgement of the difficult clinical circumstances under which the team conducted the research. It gives context to your choice of power and demonstrates pragmatism and prudence.

We entirely agree with the Reviewer that providing context is important when understanding our decision to aim for 85% power given the difficulties faced in conducting clinical research during the covid-19 pandemic. As suggested, we have added a sentence as suggested within the limitations section explaining our decision.

Thank you for providing an Appendix with the outputs of the ANOVA and mixed models. This is very helpful to contextualise the data you present. I strongly advise, in fact I think it should be mandatory, that it is included in the final manuscript or added as supplemental material.

We have included these tables within our Supplementary Information file and are grateful to the Reviewer for ensuring our statistical reporting is more robust.

I appreciate your choice of presenting placebo-corrected percentage change from baseline in your figures for readability and ease of communication to a clinical audience. However, I disagree with you with regards to omitting the full details of the statistical modelling. It is one of the CONSORT guidelines: “For each primary and secondary outcome, results for each group, and the estimated effect size and its precision (such as 95% confidence interval)”. This should really be minimum standard requirement in the reporting of RCT trials.

Please see: <https://www.goodreports.org/reporting-checklists/consort/info/#17a> for further detail.

You say: “We are keen to avoid making the text too ‘number heavy’ and believe it is important to present results in context for clarity, accepting that some readers might be unfamiliar with the specific endpoints examined”, but from my prospective ‘number heavy’ is including too many p-values of secondary/exploratory endpoints while omitting crucial outputs of your primary endpoint’s statistical modelling! I can do with fewer p-values and a greater focus on the primary endpoint. Hence my suggestion of taking p-values off the figures. Figures speak for themselves. Happy for the editorial guidelines to have final say.

We appreciate the Reviewer’s comments and understand the requirements set out in CONSORT and as above. In line with these, we have added the details of our statistical models within the Supplementary Information, including 95% confidence intervals. We agree with the Reviewer that p values are open to over-interpretation and do understand their point of view. We have not adjusted the figures but, as requested by the Editors, have reported the specific p values shown within each figure legend.

Thank you for adding details of covariates used in the methods section (again this is standard practice reporting an RCT) and of corrections for multiple comparisons (ditto). This will add scientific rigour to your paper.

We are very grateful to the Reviewer for highlighting that we had omitted this from our initial manuscript – we completely agree that this information is mandatory for transparent statistical reporting.